# Adaptive Conformal Regression with Split-Jackknife+ Scores

**Nicolas Deutschmann**
*IBM Research*[†]

**Mattia Rigotti**[*]                                                                 *mrg@zurich.ibm.com*
*IBM Research*

**María Rodríguez Martínez**
*IBM Research*[‡]

Reviewed on OpenReview: *https://openreview.net/forum?id=1fbTGC3BUD*

## Abstract

We introduce an extension of conformal predictions (CP) based on a combination of split-CP and the Jackknife+ procedure that enables tuning score functions to calibration data and designed to produce dynamically-sized prediction interval in regression settings. We motivate this method with theoretical results on distribution-dependent conditional coverage guarantees for split-CP and Jackknife+ prediction sets which are determined by the statistical dependence between input data and prediction scores. This dependence can be reduced by adapting the score function to the data distribution, thereby improving the conditional validity of conformal prediction sets. As an illustration, we construct a variant of the MADSplit conformal regression procedure where conditional mean estimates are computed in-distribution and show through empirical validation that our method is more robust to overfitting effects than the original method, while being more sample-efficient than modern ECDF-based methods.

## 1 Introduction

Developing reliable estimates of machine learning model predictions is a crucial component of making these predictions safely usable in many application domains. If model predictions are used to inform decision making, understanding predictive errors can help better understand risks in ways that raw predictions cannot. A number of approaches have been proposed to estimate potential prediction errors, based on heuristics Lakshminarayanan et al. (2017); Guo et al. (2017), bayesian modeling (Gal & Ghahramani, 2016; Kuleshov et al., 2018; Maddox et al., 2019), or frequentist approaches such as conformal predictions.

Conformal prediction (CP) (Vovk et al., 1999; Saunders et al., 1999; Vovk et al., 2005) provide a framework to perform rigorous post-hoc uncertainty quantification on machine learning predictions. CP converts the predictions of a model into prediction sets that verify guarantees on expected coverage (the probability that the realization is contained in the predicted set) with finite calibration data and no constraints on distributions. These guarantee are highly important in safety-critical applications (Lekeufack et al., 2023) or domains where enacting decisions is costly (Fannjiang et al., 2022), but come at the price of less flexibility compared to alternative approaches.

The original formulation of CP was best-suited for classification problems, and indeed the early work of Lei & Wasserman (2012) identified that a straightforward application of the formalism would predict constant-

---

[†]Now at Cradle
[*]Corresponding author
[‡]Now at Yale School of Medicine

size intervals as prediction sets, highlighting the need for methods to make these prediction intervals (PI) dynamic. A theoretical basis for conformal regression (CR) by Lei et al. (2017) was followed by multiple new approaches for CR that maintain the original guarantees of CP while also improving local behavior (Romano et al., 2019; Guan, 2020; 2022; Han et al., 2023; Lin et al., 2021).

Despite these developments proposing solutions that empirically improve the adaptivity of conformal prediction intervals, theoretical results have been established showing that several definitions for finite-sample conditional guarantees (Barber et al., 2020a; Vovk, 2012; Lei & Wasserman, 2012), so that general, well-distributed coverage seems impossible. Our work, however, shows that a weaker notion of conditional coverage guarantee is possible, but can only be established post-hoc and depends on properties of the data and conformal score distributions. This motivates the use of adaptive conformal scores, for which we propose a general method in which we use a fixed model with a held-out calibration set *à la* split-conformal and re-purpose the Jackknife+ (Barber et al., 2020b) procedure to adapt the score function to the calibration data. To this end, we slightly extend the original Jackknife+ formulation to apply to general score functions and show that our data-dependent conditional coverage guarantee extends to Jackknife+ prediction sets. We propose a practical realization of this scheme in the form of an improvement of the MADSplit method and show that this construction yields satisfying empirical results, avoiding issues of the original MADSplit when models overfit and obtaining better results than methods based on conditional score ECDF in low data regimes.

## 1.1  Definitions

Let us establish definitions that will be reused throughout this manuscript. We consider a probability space $M_{X \times y} = (\Omega_X \times \Omega_y, \mathcal{F}_X \otimes \mathcal{F}_y, \pi)$ defined over a product of measurable spaces[1], $(\Omega_X, \mathcal{F}_X)$, and $(\Omega_y, \mathcal{F}_y)$, to which we refer respectively as *input-* and *label-space*.

We work in the context of split-conformal predictions, *i.e.,* we sample a training dataset which we use to produce a predictive model $m(X) : \Omega_X \to \Omega_y$ through a training algorithm. We then further sample *i.i.d* $(X_i, y_i)_{i=1...N+1}$, also independent from the training data and $m$. We refer to the first $N$ sampled points as the calibration dataset $\mathcal{C}_N = (X_i, y_i)_{i=1...N}$ and to $(X_{N+1}, y_{N+1})$ as the test point. For conformal predictions, we consider a score function $s(X, y) : \Omega_X \times \Omega_y \to \mathbb{R}$ and label $s_i = s(X_i, y_i)$. We define the usual calibration intervals with global risk $\alpha \in [0, 1]$ as $S^\alpha(X) = \{y \in \Omega_y | s(X, y) \leq q_{1-\alpha}(s_i)\}$, where $q_{1-\alpha}(s_i)$ is the $\lceil(1-\alpha)(N+1)\rceil/N$-quantile of the empirical score distribution on $\mathcal{C}_N$, *i.e.* the $\lceil(1-\alpha)(N+1)\rceil$-th sorted score among $\{s_1, \ldots, s_N\}$.

## 1.2  Related Work

### 1.2.1  Conformal predictions

Conformal predictions were introduced by Vovk et al. (1999; 2005) and define procedure to make set predictions from single point predictors such that distribution-free, finite sample guarantees hold under a condition of exchangeability between training and test data. In recent years, many additions have been developed to better adapt the method to modern machine learning. Among these, split conformal predictions (Papadopoulos et al., 2002) rely on a single, independently-trained predictive model and a held-out calibration set $\mathcal{C}_N$. This is the formalism used to define predictions sets $S^\alpha$ in Section 1.2.

In split conformal predictions, under the condition that the calibration data $\mathcal{C}_N$ is exchangeable with the test point $X_{N+1}, y_{N+1}$, a global coverage guarantee can be established independently of the data probability function $\pi$, which is valid marginally over $\mathcal{C}_N$.

**Theorem 1** (Global Split-Conformal Coverage Guarantee (Papadopoulos et al., 2002))

$$\mathbb{P}(y_{N+1} \in S^\alpha(X_{N+1})) \geq 1 - \alpha.$$

---

[1]It goes without saying that the probability distribution $\pi$ is not a product measure, *i.e.* $y \not\perp X$.

A pedagogical review of CP methods, which the authors have found useful, can be found in Angelopoulos & Bates (2021).

### 1.2.2 Conditional coverage in conformal predictions

A limitation of Theorem 1 is that it does not offer conditional guarantees on sub-regions of $\Omega_X$, so that undesirable behavior is still possible, such as an uneven coverage. This can result in coverage that is unfairly distributed across groups, labels, or generally concentrated in some data space regions. While satisfying solutions for finite numbers of groups and labels exist (Vovk, 2012; Lu et al., 2022), there are impossibility theorems strongly constraining more general notions of coverage (Barber et al., 2020a; Vovk, 2012; Lei & Wasserman, 2012). Barber et al. (2020a) introduce two such definitions:

**Definition 1** (Distribution-free conditional coverage (Barber et al., 2020a))
*A conformal prediction procedure with prediction sets $S^\alpha (X_{N+1})$ verifies distribution-free conditional (DFC) coverage if $\forall x \in \Omega_X$,*

$$\mathbb{P}\left(y_{N+1} \in S^\alpha (X_{N+1}) \,\middle|\, X_{N+1} = x\right) \geq 1 - \alpha,$$

*for any data distribution $\pi$.*

**Definition 2** (Approximate distribution-free conditional coverage (Barber et al., 2020a))
*A conformal prediction procedure with prediction sets $S^\alpha (X_{N+1})$ verifies approximate distribution-free conditional (ADFC) coverage if for any $\omega_X \subset \Omega_X$ such that $\mathbb{P}(x \in \omega_X) \geq \delta$,*

$$\mathbb{P}\left(y_{N+1} \in S^\alpha (X_{N+1}) \,\middle|\, X_{N+1} \in \omega_X\right) \geq 1 - \alpha,$$

*for any data distribution $\pi$.*

Results from Vovk (2012); Barber et al. (2020a) imply the impossibility of DFC coverage and Barber et al. (2020a) show the impossibility of ADFC coverage with finite sample guarantees.

Despite these results, there is a growing body of work on improving the local properties of conformal predictions and especially conformal prediction intervals (PI), usually combining theoretically proven global bounds and empirically established local or conditional properties. Three major types of approaches have been developed with this goal, which we describe below.

### 1.2.3 Reweighted scores

The earliest efforts on adaptive CR proposes to rescale the usual nonconformity scores (Papadopoulos et al., 2008; Lei et al., 2017) with local information. The most prominent approach is the MADSPLIT approach (Lei et al., 2017), which uses $s(\hat{y}(X), y) = |\hat{y}(X) - y| / \hat{\sigma}(X)$, where $\sigma(X)$ is an estimate of the conditional score mean $\mathbb{E}[|\hat{y}(X) - y| | X]$. Importantly in comparison with our proposed method, in MADSPLIT this estimate is performed on the training split, which preserves CP coverage guarantees, but yields poor performance when errors differ between training and test sets, which can happen due to overfitting or distribution shifts.

### 1.2.4 Model-based interval proposals

A second type of approach is based on choosing machine-learning models that themselves encode some notion of uncertainty. The canonical formulation is conformal quantile regression (Romano et al., 2019), although other approaches based on predicting distribution parameters exist. While attractive in principle, a limitation of these methods is that poor modeling performance immediately leads to poor PI. This makes PI for difficult-to-predict examples the least trustworthy, despite often being the most important.

### 1.2.5 Reweighted score distributions

The most recent type of adaptive conformal regression methods is based on computing adaptive distortions of the non-conformity score empirical cumulative distribution function (ECDF), proposed originally by Guan (2020). Given a test point, the ECDF is modified by giving a similarity-based weight to each calibration point,

yielding an estimate of the test-point-conditional score ECDF. If the ECDF estimator is good, the result is perfect local coverage and adaptivity. Multiple variants have been proposed to optimize computational complexity or ECDF estimation, including LCR (Guan, 2020), LVD (Lin et al., 2021), SLCP (Han et al., 2023). Despite only verifying global coverage guarantees with finite samples, SLCP also was proved to achieve DFC coverage asymptotically. With finite samples, however, estimating the high quantiles of the conditional CDF can require significant statistics, as many calibration points are needed *near every test point.* Furthermore, as we discuss in Section 5, the proposed localization-weight tuning methods often yield poor results on complex, high-dimensional data.

## 2  Local Coverage and Score-Input Independence

We begin by defining and characterizing the goals of adaptive conformal regression in terms of guarantees and tightness. This analysis is general and extends beyond the method we propose, but will also provide motivation for our approach and a framework to understand its applicability.

Informally, the objective of PI prediction to achieve two desirable properties: local coverage guarantees for any input point $X$ and interval bounds that perfectly adapt to capture the label distribution.

### 2.1  Local Coverage Guarantees

We introduce a further weakened notion of conditional coverage compared to ADFC and DFC coverage, abandoning distribution-independence:

**Definition 3** (Approximate $\pi$-dependent conditional coverage)
*A conformal prediction procedure with prediction sets $S^\alpha(X_{N+1})$ verifies approximate $\pi$-dependent conditional ($A\pi DC$) coverage for a given probability function $\pi$ if there is an $\alpha' \in [0, 1)$ such that for any $\omega_X \subset \Omega_X$ where $\mathbb{P}(x \in \omega_X) \geq \delta$,*

$$\mathbb{P}\left(y_{N+1} \in S^\alpha(X_{N+1}) \,\middle|\, X_{N+1} \in \omega_X\right) \geq 1 - \alpha',$$

The formulation of Definition 3 is amenable to a sufficient condition to ensure strong conditional coverage for conformal intervals defined as in Section 1.1.

**Proposition 1** (Sufficient condition for $A\pi DC$)
*Given a data distribution $\pi$, if $S^\alpha(X)$ is defined from a score $s(X, y)$ such that*

$$X \perp s(X, y)$$

*then $A\pi DC$ is verified with $\alpha' = \alpha$ and $\delta = 0$.*

Indeed, if the score distribution is input-independent, then its conditional quantiles are as well, so that estimating quantiles globally is the same as estimating them locally. This has been observed in previous work and is tightly related to the definition of the orthogonal loss in Feldman et al. (2021), which measures the correlation between the local coverage and the local interval size.

Of course obtaining an independent score function is impossible in general, but it helps understanding the origin of the following guarantee, which can be used to guarantee $A\pi C$ coverage for certain problems.

**Proposition 2** (Bound on conditional coverage)
*If the mutual information between $X$ and $s(X, y)$, $MI(X, s)$ is finite, for any $\omega_X \in \mathcal{F}_X$ such that $\mathbb{P}(X \in \omega_X) > \delta$, then*

$$\mathbb{P}(y \in S^\alpha(X) | X \in \omega_X) \geq (1 - \tilde{\alpha}), \qquad \text{where } \tilde{\alpha} = \alpha + \sqrt{1 - e^{-MI(X,s)}}/\delta.$$

In plain words, conformal prediction sets verifying a global coverage guarantee with a risk $\alpha$ can verify $A\pi DC$ where the guarantee $1 - \tilde{\alpha}$ is the global coverage guarantee $1 - \alpha$ degraded by a penalty term $\sqrt{1 - e^{MI(X,s)}}/\delta$

This follows from Lemma 1 and is proven in the supp. material. The bound is vacuous for low-probability sets, an issue that we argue is inherent to this type of results (see Appendix F.1).

Unlike global conformal guarantees which can be specified *a priori, Proposition 2 provides a distribution- and score-function-dependent conditional coverage guarantee.* Nevertheless, given a probability function $\pi$, a score function $s$ yielding a sufficiently small penalty coefficient $\sqrt{1 - \exp\left(-\mathrm{MI}(X, s)\right)}$ can provide conditional guarantees. As we discuss in Section 4.1, this helps understand how the rescaled score used by MADSPLIT and our own method achieve adaptive intervals.

While the conditional risk $\tilde{\alpha}$ is problem-dependent, Proposition 2 is still a result *about* conformal predictions in that it characterizes their conditional properties and applies generally for finite samples and without making hypotheses about classes of distributions.

## 2.2 Measuring the Adaptivity of PI

In the context of regression, all CP methods we are aware of predict intervals(*i.e.* error bars), creating a tight connection between coverage and the usual absolute error metric used in regression. Indeed, if a model produces a prediction $m(X)$ for a sample $(X, y)$, a typical CR prediction interval will take the form $I(X) = [m(X) - l(X), m(X) + u(X)]$, so that the value $y$ is covered if $\mathrm{AE}(X, y) = |y - m(X)| \leq \max(u(X), l(X))$. For symmetric methods such as our own and the methods from literature we compare to, we further have $u(X) = l(X)$ so that coverage is equivalent to $\mathrm{AE}(X, y) \leq |I(X)|/2$. Conditional coverage with risk $\alpha$ is obtained if this inequality is verified with a conditional probability of $1 - \alpha$ (given $X$). We can rephrase this statement in terms of the conditional absolute error quantile $q_{1-\alpha}\left(\mathrm{AE}(X, y)|X\right)$ as follows:

$$q_{1-\alpha}\left(\mathrm{AE}(X, y)|X\right) = \frac{|I(X)|}{2}. \tag{1}$$

Of course, the results of (Vovk, 2012; Lei & Wasserman, 2012) state that no CP method can achieve this goal generally and exactly, but approximating this relationship on practical application yields real benefits: such error bars can be trusted to cover most points even in high-error or high-noise input regions while avoiding being over-conservative.

It is common to evaluate adaptive CR methods based on two metrics computed on held-out test data: the coverage rate (Cov.) and the mean PI size (IS), under the understanding that, at fixed coverage, smaller intervals means tighter correlation between PI size and error rate. However, as we show in Appendix D, an oracle predicting ideal intervals might in fact *increase* the mean interval size compared to a non-adaptive approach. We therefore propose several *new metrics* to evaluate adaptivity of PI prediction methods which capture more fine-grained information about the alignment between interval sizes and errors. In all cases, we measure errors in regression with the absolute error score.

### 2.2.1 Score-Interval Size correlation $\tau_{\mathsf{SI}}$

A natural metric to measure whether absolute errors and interval sizes have a monotonous relationship (higher errors map to larger intervals) is to use a rank correlation metric and we therefore define $\tau_{\mathrm{SI}}$ as the Kendall-$\tau$ rank score between these raw measures. A downside of this metric is that it is sensitive to problem-dependent score distributions. Note that we refer to absolute errors are scores in this case for generality, as this metric could be applied to other notions.

### 2.2.2 Score-Interval Quantile correlation $\tau_{\mathsf{SIQ}}$

While evaluating the relationship in Eq. (1) is impractical, we can marginalize over inputs $X$ that approximately share the same interval size $|I(X)|$, which we obtain by grouping points by quantiles.

We group test samples by their predicted interval size quantiles (deciles in our experiments) and compute absolute error score averages for each quantile. The rank correlation between the quantile mid-point value and these averages, $\tau_{\mathrm{SIQ}}$, captures similar information as $\tau_{\mathrm{SI}}$ but gets closer to properly testing Eq. (1).

### 2.2.3 Score-Quantile vs Interval Quantile determination

In order to properly verify the Eq. (1), averaged over approximately-identical interval size samples, we turn to the coefficient of determination $R^2$.

For a given $d$-decile ($d = 1, \ldots, 10$), with a left-edge value $I_d$, we ideally have $1 - \alpha$ coverage, which is obtained if $I_d/2$ is approximately equal to the $1 - \alpha$-error-score quantile in this interval-size quantile, $q_{1-\alpha}(\mathrm{AE}(\mathrm{X})|I_d \leq I(X) \leq I_{d+1})$. We therefore expect

$$q_{1-\alpha}(\mathrm{AE}(\mathrm{X})|I_d \leq I(X) \leq I_{d+1}) \simeq \frac{I_d}{2}. \tag{2}$$

We introduce $R^2_{\mathrm{SQI}}$, the coefficient of determination verifying this relationship. A high value of $R^2_{\mathrm{SQI}}$ indicates that higher interval sizes are neither too large or too small in each quantile.

## 3 Approaching Conditional Coverage with Adaptive Scores

As we show in Section 2, a problem-dependent notion of conditional coverage can be guaranteed based on the statistical dependence between input-space data $X$ and the conformal score $s(X, y)$. We interpret this distribution dependence, manifest in the mutual information penalty term of Proposition 2, as a sign that information about the data distribution needs to be injected into the conformal calibration procedure to obtain conditional coverage guarantees.

Learning parameters of score functions from data is not entirely new and indeed C-ECDF-based approach learn their conditional distributions while MADSplit relies on an estimator of the conditional error mean. Both approaches rely on kernels to give local weights to the data used for this estimates but neither provides a satisfying mechanism to tune the hyperparameters of these kernels and rely on heuristics, which we empricaly show to fail in certain experiments. Furthermore, the conditional mean score estimates of MADSplit are obtained from training data which can have significantly different errors from test data due to overfitting.

This motivates the need for a more systematic approach to learn the parameters of adaptive conformal score functions, which we formulate below.

### 3.1 Learning scores on calibration data: Split-Jackknife+ Calibration

We propose a new conformal calibration approach that combines elements of split-conformal and Jackknife+ conformal predictions (Barber et al., 2020b): like with split-conformal predictions, the predictive model is trained once on a separate training set and conformal calibration is performed on a held-out calibration dataset. Unlike split-conformal predictions however, we wish to introduce information from the calibration set in the definition of the score function, crucially taking care of not breaking exchangeability.

The Jackknife+ procedure was originally introduced to allow a leave-one-out approach to conformal calibration, training $N$ predictive models with a single sample left out, which is then used to compute a calibration score value. In our approach, the same procedure is applied to train $N$ score functions evaluated on the output of a single pre-trained predictive model.

We consider a family of score functions $s(X, y; \theta)$ with parameters $\theta$ and construct score matrix $s_{ij} = s(X_i, y_i; \theta_{ij})$, indexed by the union of the calibration set and the test point, $i, j \in J^+ = [1, \ldots, N+1]$. Each trained parameter $\theta_{ij}$ is obtained by applying a learning algorithm $A$ to a restricted training set $T_{ij} = \{(X_k, y_k)|k \in J^+/\{i, j\}\}$. Of this formal construction, we can generally construct only $N$ entries without access to the test label $y_{N+1}$ and the Jackknife+ construction relies only on these accessible entries, $s_{i(N+1)}$.

At test-time, we exploit the symmetry of the learned parameters $\theta_{ij}$ to define the prediction set $C_\alpha(X_{N+1})$:

$$C_\alpha(X_{N+1}) = \left\{ y \in \Omega_y \,\middle|\, q_\alpha\left(\frac{s_{i(N+1)}}{s(X_{N+1}, y; \theta_{i(N+1)})}\right) \geq 1 \right\}. \tag{3}$$

This formulation is an extension of the original formulation in (Barber et al., 2020b) which was limited to conformal regression with an absolute error score and allows for a generalized applicability of the method. In this manuscript, we consider the task of adapting score function, but equation 3 can be used for the original intended goal of Jackknife+ if the learned parameters $\theta_{ij}$ are those of the predictive model.

As we prove in Appendix E, this formulation extends the global coverage guarantee of the Jackknife+ method:

**Proposition 3**
*Consider the data given in Section 1.1 and the prediction intervals defined in Eq. (3), then, marginally over the calibration set,*

$$\mathbb{P}\left(y_{N+1} \in C_\alpha\left(X_{N+1}\right)\right) \geq 1 - 2\alpha.$$

**Proposition 4**
*Consider the data in Section 1.1, the score matrix $s_{ij}$ defined above and define $k_\alpha = \lceil(1-\alpha)(N+1)\rceil$. We define the score ratio $\sigma$ as the $k_\alpha$-th among the sorted score ratios $s_{(N+1)i}/s_{i(N+1)}$. Then for any $\omega_X \in \mathcal{F}_X$ such that $\mathbb{P}(X \in \omega_X) \geq \delta$*

$$\mathbb{P}\left(y_{N+1} \in C_\alpha(X_{N+1})|X \in \omega_X\right) \geq 1 - \alpha", \qquad where\ \alpha" = 2\alpha + \sqrt{1 - e^{-\text{MI}(X,\sigma)}}/\delta.$$

Note that this result applies also to the original Jackknife+ approach where the model, rather than the score, is trained $N$ times.

Proposition 4 provides a way to estimate conditional mis-coverage risk given a Split-Jackknife+ procedure with global risk $\alpha$ very similar to that of Proposition 2. However, because the Split-Jackknife+ procedure allows the optimization of the score function, this also opens the door to a principled way of optimizing for improved local guarantees, as we discuss below.

### 3.2 Optimizing scores with a mutual-information objective

When a score function has free parameters to optimize, such as kernel parameters, there is currently no satisfying loss function to guide parameter choice. The penalty term in Proposition 2 provides a well-motivated target to minimize in order to provide stricter conditional coverage guarantees. We therefore propose the following score loss:

$$\mathcal{L}_{\text{score}} = \text{MI}\left(\epsilon(X), s\left(X, y; \theta\right)\right), \tag{4}$$

where $\epsilon$ is an embedding function (possibly the identity). The use of $\epsilon$ is motivated by practical considerations in modern deep learning applications where inputs can be high-dimensional, making mutual information estimation challenging. As long as the score can be re-cast as a function of $\epsilon(X)^2$, minimizing $\mathcal{L}_{\text{score}}$ improves a notion of conditional coverage: Proposition 2 would apply on subset of the embedding space $\epsilon(\Omega_X)$ – which is also be more meaningful for deep learning applications where the data manifold represents a sliver of the full space $\Omega_X$.

## 4 Method: Split-Jackknife+ Rescaled Scores

In this section, we provide a concrete realization of the Split-Jackknife+ adaptive score procedure for regression problems, in the form of an improvement over the original MADSplit method. We use the same conditional-mean-rescaled absolute error score but propose two changes: we use our procedure to perform in-distribution estimates of the conditional mean, and if kernels need to be tuned, we use $\mathcal{L}_{\text{score}}$ to choose their parameters.

---

[2] such as if $s(X, y)$ is a score function evaluated on $\mu(X)$ and $y$, where $\mu = \nu \circ \epsilon$ is a neural network with a hidden layer described by $\epsilon$

### 4.1 Split-Jackknife+ Rescaled Scores

We first formulate the method assuming that we have a fixed kernel function $K(X, X) \to \mathbb{R}_+^*$. Our base score function for regression is the usual absolute error score $s(X, y) = |\mu(X) - y|$, where $\mu$ is our pre-trained model independent from the calibration and test data. We define Jackknife+ rescaled scores as

$$s_{ij}^+ = \frac{s(X_i, y_i)}{\hat{\bar{s}}_{ij}(X_i)}, \tag{5}$$

where $\hat{\bar{s}}_{ij}(X_i)$ is a Nadaraya-Watson estimator of the conditional mean of $s$ at $X_i$ based on $K$ and using $T_{ij}$ as a training set:

$$\hat{\bar{s}}_{ij}(X_i) = \sum_{k \in J^+/\{ij\}} \frac{K(X_i, X_k)s(X_k, y_k)}{Z_{ij}} \qquad \text{with} \qquad Z_{ij} = \sum_{k \in J^+/\{ij\}} K(X_i, X_k). \tag{6}$$

As described in Section 2.1, the actual calibration scores are $s_i^+ = s_{i(N+1)}^+$ for $i = 1, \ldots, N$, and their formulation as a ratio permits a more explicit definition of the prediciton sets as intervals around the model predictions:

$$C_\alpha(X_{N+1}) = \left[ \mu(X_{N+1}) \pm q_\alpha\left( \left\{ \hat{\bar{s}}_{(N+1)i} s_i^+ \right\} \right) \right], \tag{7}$$

where the calibration scores are rescaled by the estimate of the conditional score mean around the test point $X_{N+1}$. We provide an explicit description of the calibration and inference procedure in Algorithm 1.

From a formal point of view, rescaling scores even with a perfect conditional mean predictor is not guaranteed to improve the adaptability of prediction intervals except if the conditional score distribution of $s$ given $X$ belongs to a scale family such that changes in conditional mean entirely determine changes in conditional quantiles. Nevertheless, intuitively rescaled scores should capture some of the conditional dependence of the quantiles and therefore improve adaptability. We confirm this intuition empirically in REF by showing that rescaled scores yield a significant reduction in the score-input mutual information compared to the raw absolute error score, which translates to the empirical performance of the method described in Table 1.

## 5 Experimental Evaluation

Throughout this section, we fix the kernel for our method to a simple (approximate (PyNNDescent; Dong et al., 2011)) K-nearest-neighbor (KNN) kernel ($K_{ij} = 1$ if $j$ is among the KNN of $i$, otherwise $K_{ij} = 0$), setting $K = 10$. We've found that this approach yields competitive results in many cases without any tuning for our approach. We discuss the potential gains of kernel tuning in Appendix C.2.

### 5.1 Comparing Fairly to Previous Work

Tuning kernels is a key element of all existing post-hoc adaptive PI and should, to some extent, be considered part of the method. Nevertheless, to perform fair comparisons, we have tried to strike a balance between considering PI proposal methods as end-to-end and evaluating the CP method in isolation by making best effort extensions.

We evaluate our method against MADSPLIT (Lei et al., 2017) and LVD (Lin et al., 2021) as representatives of the two classes of post-hoc CR, which both rely on RBF kernels. MADSPLIT proposes setting the bandwidth parameter to the median of the distance distribution in the model-training data, but we found this to rarely be optimal and we therefore also used the same KNN kernel as for our method. LVD optimizes an anisotropic RBF $K(x, y) = \exp(-|A(x - y)|^2)$ where $A$ is a matrix on the same task as the predictive model, which lead to a large fraction of infinite intervals[3]. A large number of neighbors would be required

---

[3]LVD can predict infinite intervals by design: it estimates quantiles of the conditional score distribution around a test input $X$ by reweighting the empirical calibration distribution. If the reweighted score distribution has very low entropy, the effective sample size is very reduced, so that the estimated quantile is infinite. In a high-dimensional setting, it seems to often the case that the effective sample size obtained when learning a predictive model (estimating the conditional mean) is too small.

for a KNN kernel for LVD so we instead rescale $A \to \lambda A$ and tune $\lambda$ until the fraction of infinite intervals is negligible. Note that we also attempted to evaluate SLCP (Han et al., 2023), but were unable to tune kernels to reach comparable metrics to the other methods, and we therefore left it out of our results.

## 5.2 One-Dimensional Regression

Let us start by applying our methods to simple regression problems that will help illustrate how our approach operates. We define the following random variables

$$y = f_0 + X^2 \sin\left(k_f X + \phi_f\right) + \epsilon \text{ and } \nu(X) = \epsilon_0 + |\sin\left(k_\epsilon X + \phi_\epsilon\right)|, \quad \text{with } X \sim \mathrm{U}(0,1), \quad \epsilon \sim \mathcal{N}(0, \nu(X)).$$

While serving an illustrative purpose, this relationship is designed to display heteroscedatic noise whose $X$-dependency is decoupled from the conditional expectation of $y$ given $X$.

We use 1000 calibration and 10000 test points to evaluate our method on a random-forest regressor trained on independent data and indeed find intervals that achieve the target global coverage and dynamic interval sizes, as show in 1.

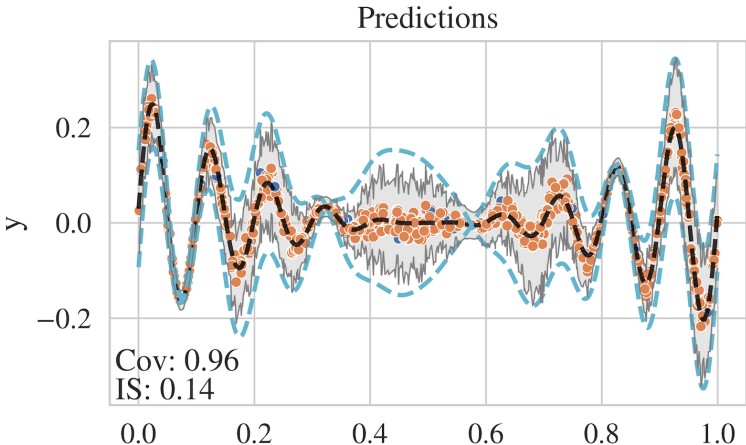

Figure 1: Evaluation of our approach on a 1D regression problem. The dashed black line is the expected value of $y$ given $x$ and the dashed blue line shows an interval of two standard deviations around that expected value. Orange points correspond to model predictions and the gray band display prediction intervals.

This simple setting already allows us to compare performance with other adaptive PI methods. We display the calibration-set-size dependence of PI metrics in Fig. 2.

These results highlight a more general trend: our method is more robust than LVD in the low data regime but tends to be similar with enough data. MADSplit is close to our method when noise is high, but if overfitting becomese visible, our method avoids the bias felt by MADSplit.

## 5.3 Results on complex data

We consider four datasets in addition to our 1D regression problem. In all cases, we fix the training data and train an adapted regression model by optimizing the mean squared error. Details about data processing, embeddings used for localisation and predictive models can be found in Appendix B. Prediction intervals are predicted with low and high amouts of calibration data over 10 repetitions where calibration and test data are resampled from the held-out data. The tasks are:

**Two TDC molecule property datasets (Huang et al., 2021):** drug solubility and clearance prediction from respectively 4.2k and 1.1k SMILES sequences[4].

---

[4]string-based description of molecular structures (Weininger, 1988).

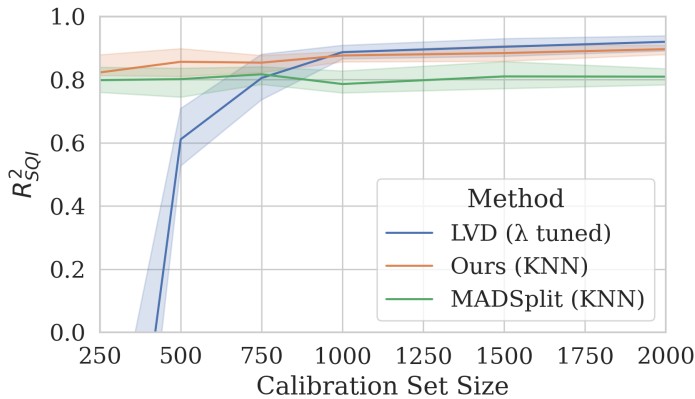

Figure 2: Calibration-set-size dependence of PI metrics evaluated on an independent test set.

**The AlphaSeq Antibody binding dataset (Engelhart et al., 2022):** binding score prediction for a SARS-CoV-2 target peptide from 71k amino-acid sequences.

**Regression-MNIST (LeCun & Cortes, 2005):** Regression of the numeric label of MNIST images with test-time augmentations (*e.g.* predicting the number "9.0" when presented with a picture of the digit "9"). This task was selected due to its irregular error distribution, which is expected to challenge our rescaling approach by breaking even approximate relationships between conditional means and quantiles, see Appendix C.4.

We detail our results in Table 1. In summary, we confirm that our method outperforms LVD when data is scarce, while it is usually comparable with MADSPLIT with our performance improving faster with growing sample size. The results on sequence data with higher statistics confirm that our method is competitive with both methods. Interestingly, while AlphaSeq has the most data, our method dominates others. This is due to a combination of increased overfitting and scattered data distribution as discussed in Appendix C.3. Finally, the MNIST task is peculiar: all methods have comparable performance as measured by correlation measures but only LVD has both tight and adaptive interval sizes, as measured by $R^2_{SQI}$. We attribute this to the irregular error distributions due to possible number confusions, which leads to highly variable conditional mean/quantile ratios, as shown in Appendix C.4.

### 5.3.1 Conditional Coverage Bounds

We complement the results of Table 1 with estimates of the data-dependent coverage bounds from Proposition 2 for raw absolute error scores and MADSplit, and Proposition 4 for our method. As we discuss in Section 3.2, computing mutual information in high-dimensional spaces is challenging and we therefore evaluate the mutual information between scores and mapped inputs in the latent spaces of our models, so that the bounds are valid in this space. We rely on an approximation by using a PCA to further reduce dimensionality, exploiting the tendency of regression models to organize latent representation linearly: we found that a very large fraction of the variance can be expressed with only 4 components, which greatly simplifies MI estimation. The estimation is performed with MINE (Belghazi et al., 2018) as non-parametric method like the KSG estimator (Kraskov et al., 2004) yielded poor results above one or two dimensions. As we show in Table 2, the empirical conditional bounds are rather weak and most likely not tight as the ordering between MADSplit and our method is reversed compared to other performance metrics in Table 1.

## 6 Conclusions

Our empirical results show that our new approach provides a satisfying solution to the weaknesses of existing post-hoc adaptive conformal regression methods. On the one hand, our use of the Jackknife+ procedure to

| Dataset | $N_{\text{cal}}$ | Method | Cov. (≳ 0.95) | IS ↓ | $R^2_{SQI}$ ↑ | $\tau_{\text{SIQ}}$ ↑ | $\tau_{\text{SI}}$ ↑ |
|---|---|---|---|---|---|---|---|
| 1D | 500 | Ours | 0.96(1) | **0.18(1)** | **0.87(3)** | 0.91(4) | **0.35(1)** |
| | | MADSPLIT | 0.952(9) | **0.18(1)** | 0.80(4) | **0.96(3)** | **0.344(9)** |
| | | LVD | 0.982(3) | 0.21(1) | 0.61(9) | 0.86(4) | 0.33(1) |
| | 2000 | Ours | 0.952(2) | **0.171(5)** | **0.90(2)** | 0.96(4) | **0.36(1)** |
| | | MADSPLIT | 0.952(9) | 0.18(1) | 0.80(4) | 0.96(3) | 0.344(9) |
| | | LVD | 0.968(3) | **0.174(3)** | **0.92(2)** | 0.96(3) | **0.361(6)** |
| TDC Sol. | 400 | Ours | 0.95(1) | 5.3(5) | 0.4(1) | 0.6(1) | 0.09(1) |
| | | MADSPLIT | 0.95(1) | 5.0(3) | **0.53(5)** | **0.71(6)** | **0.182(3)** |
| | | LVD | 0.958(7) | 5.2(3) | 0.3(5) | **0.7(1)** | 0.07(5) |
| | 800 | Ours | 0.96(1) | **5.2(4)** | 0.5(1) | 0.7(1) | 0.11(1) |
| | | MADSPLIT | 0.951(6) | **5.1(2)** | **0.58(7)** | 0.71(5) | **0.179(6)** |
| | | LVD | 0.960(9) | 5.4(2) | 0.4(3) | 0.77(7) | 0.07(1) |
| TDC Clear. | 60 | Ours | 0.95(2) | **$1.9(3)\cdot 10^2$** | **0.2(3)** | **0.6(1)** | **0.37(3)** |
| | | MADSPLIT | 0.95(2) | **$1.9(5)\cdot 10^2$** | **0.2(3)** | **0.60(9)** | **0.39(1)** |
| | | LVD | 0.97(2) | $2.2(2)\cdot 10^2$ | ≪ 0 | −0.3(6) | 0.1(2) |
| | 240 | Ours | 0.96(1) | **$1.65(9)\cdot 10^2$** | **0.3(2)** | **0.5(1)** | **0.39(7)** |
| | | MADSPLIT | 0.96(1) | **$1.64(8)\cdot 10^2$** | **0.41(5)** | **0.6(1)** | **0.41(7)** |
| | | LVD | 0.97(1) | $2.02(5)\cdot 10^2$ | ≪ 0 | 0.1(2) | 0.1(1) |
| $\alpha$Seq CoVID | 2000 | Ours | 0.952(6) | **4.5(1)** | **0.24(8)** | **0.5(1)** | **0.05(1)** |
| | | MADSPLIT | 0.952(5) | **4.6(1)** | **0.1(2)** | 0.2(2) | 0.01(2) |
| | | LVD) | 0.994(1) | 7.6(5) | ≪ 0 | 0.1(1) | −0.004(6) |
| | 10000 | Ours | 0.953(4) | **4.54(6)** | **0.28(6)** | **0.6(1)** | **0.052(8)** |
| | | MADSPLIT | 0.952(2) | **4.59(6)** | 0.0(1) | 0.2(2) | 0.01(1) |
| | | LVD | 0.973(4) | 5.49(4) | −0.26(4) | 0.5(1) | **0.052(8)** |
| MNIST | 2000 | Ours | 0.952(6) | **0.62(3)** | ≪ 0 | **0.90(5)** | 0.20(1) |
| | | MADSPLIT | 0.951(7) | **0.63(5)** | ≪ 0 | 0.83(8) | 0.17(1) |
| | | LVD | 0.988(4) | 4.1(5) | **−0.19(2)** | **0.86(5)** | **0.21(1)** |
| | 5000 | Ours | 0.950(3) | **0.601(8)** | ≪ 0 | **0.92(5)** | **0.23(1)** |
| | | MADSPLIT | 0.947(4) | 0.62(1) | ≪ 0 | 0.85(7) | 0.193(6) |
| | | LVD | 0.984(2) | 4.4(2) | **−0.17(3)** | **0.92(3)** | **0.233(9)** |

Table 1: Numbers in parentheses are the uncertainty on the last significant digit evaluated over 10 random samples of test and calibration data. We report $R^2_{SQI} \ll 0$ if it is significantly lower than $-2$. Metrics where lower (higher) values are indicated with a ↓ (↑) symbol and bold entries are significantly outperforming non-bold entries while non-significantly dominating each other.

tune and evaluate the conditional mean of the non-conformity score solves the issue of MADSPLIT when evaluated on models with significant score distribution shifts between model-training and CP calibration data. On the other hand, while our method only guarantees local coverage asymptotically under rather stringent hypotheses, our empirical results show clear value compared to the diminished statistical power compared to ECDF-reweighting methods, especially in the low-calibration-data regime.

More generally, the theoretical results and data-dependent score function tuning proposed in this work provide avenues for new conformal prediction methods based on adapting score functions to data distribution properties and we hope that more refined approaches can be proposed to predict highly adaptive uncertainty sets.

| Dataset | Method | MI | Penalty | Bound |
|---------|--------|-----|---------|-------|
| 1D | AE | 0.3 | 0.5 | $\leq 0$ |
| | Ours | 0.06 | 0.3 | 0.1 |
| | MADSplit | 0.03 | 0.2 | 0.4 |
| TDC (Sol.) | AE | 0.07 | 0.3 | 0.06 |
| | Ours | 0.02 | 0.1 | 0.5 |
| | MADSplit | 0.04 | 0.2 | 0.3 |
| $\alpha$Seq | AE | 0.01 | 0.12 | 0.6 |
| | Ours | $3 \cdot 10^{-3}$ | 0.05 | 0.8 |
| | MADSplit | $8 \cdot 10^{-3}$ | 0.09 | 0.65 |

Table 2: Conditional coverage guarantees for benchmark tasks with our method, raw absolute error scores and MADSplit. The penalty is the coefficient of the penalty in Proposition 2 and Proposition 4 and the bound is given for $\delta = 0.3$.

From a theoretical point of view, we also hope to see further improvements: while Proposition 2 points to score-input dependence as an optimization target for conditional coverage, which we deem an interesting insight, we do not expect it to be optimally tight. This is not only because mutual information replaces the unwieldy total variation term of Proposition 6 for practical reasons of computability, but also because the measure-theoretic formulation is probably more general than needed for practical purposes, and a genuine notion of *local* coverage might exploit regularity properties of conformal regression scores. We therefore look forward to future work exploring how score and data distributions interplay with coverage locally, rather than conditionally.

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

## A    Algorithms

In this section, we describe the calibration and PI prediction procedures for two versions of our method in the form of pseudocode. The simplest is Algorithm 1, where the kernel is assumed to be fixed. Indeed we've found in practice that using a KNN kernel with $K = 10$ is an efficient and performant approach. We however also define Algorithm 2, where we tune $N$ kernels on calibration data, with an objective motivated by the bound of Proposition 4. For high-dimensional data, such as when using embeddings to compute kernel similarities, we use a low-dimensional PCA and compute the sum of the marginal mutual information between the score and principal component. If a PCA is not used, this sum is an upper bound on the multi-dimensional mutual information, but we've also found that using 2-4 principal components captures an overwhelming majority of the data variance for our experiments when using latent space embeddings, and yields good empirical improvement when used for kernel tuning.

---

**Algorithm 1** Jackknife+ rescaled score conformal regression without kernel tuning

**Require:**
1: Exchangeable data $\{(X_i, y_i) \in \Omega_X \times \mathbb{R} \mid i \in [-T, \ldots, N+1]\}$.
2: A learning algorithm $\mathcal{A}$ mapping a sample of $(X, y)$ pairs to a model $m(X)$.
3: A kernel $K : \Omega_X \times \Omega_X \to \mathbb{R}^+$.
4: A risk threshold $\alpha \in [0, 1]$.
5:
6: **procedure** PREDICTOR TRAINING
7:     $m = \mathcal{A}\left(\{(X_i, y_i)\}_{i \in [-T, \ldots, 0]}\right)$
8: **end procedure**
9: *All free indices below $(i, j, k)$ span $[1, \ldots, N]$.*
10: **procedure** CALIBRATION
11:     SET $K_{ij} = K(X_i, X_j)$
12:     SET $p_{ij} = K_{ij} / \sum_{k \neq i} K_{ik}$
13:     SET $s_i = |y_i - m(X_i)|$
14:     SET $s_i^+ = s_i / \sum_{k \neq i} p_{ik} s_k$
15: **end procedure**
16: **procedure** INTERVAL PREDICTION
17:     SET $Q = \lceil (1-\alpha)(N+1) \rceil$
18:     SET $p_{i(N+1)} = K_{i(N+1)} / \sum_{k \neq i, N+1} K_{ik}$
19:     SET $\hat{\bar{s}}_{(N+1)i} = \sum_{k \neq i} p_{k(N+1)} s_k$
20:     SORT $S = [\hat{\bar{s}}_{(N+1)1} s_1^+, \ldots, \hat{\bar{s}}_{(N+1)N} s_N^+]$
21:     SET $\Delta y_{N+1} = S[Q]$
22: **end procedure**
**Ensure:** Test prediction interval $[m(X) - \Delta y_{N+1}, m(X) + \Delta y_{N+1}]$.

---

---

**Algorithm 2** Jackknife+ with Kernel Tuning

---

1: **procedure** CALIBRATION WITH KERNEL TUNING($n_{\text{PCA}}$, $n_{\text{sample}}$, $n_{\text{scan}}$, $\beta_{\text{expand}} \geq 1$)
2:     SAMPLE $n_{\text{sample}}$ pairs of non-identical training inputs $(X_a, X_a')$
3:     SET $d_{\min}$, $d_{\max}$ to the extrema of $\|X_a - X_a'\|$
4:     SET $\lambda_n$ as $n_{\text{scan}}$ values evenly space in logarithmic scale in $[d_{\min}/\beta_{\text{expand}}, d_{\max} \times \beta_{\text{expand}}]$
5:     DEFINE $N$ RBF kernels $K_m$ with length scales $l_m$.
6:     **for** each $\lambda_n$ **do**
7:         **for** $m \in [1, \ldots, N]$ **do**
8:             SET $l_m = \lambda_n$
9:             SET $K_{ij;m} = K_m(X_i, X_j)$
10:             SET $p_{ij;m} = K_{ij;m}/\sum_{k \neq i,m} K_{ik;m}$
11:             SET $s_i = |y_i - m(X_i)|$
12:             SET $s_{i;m}^+ = s_i/\sum_{k \neq i,m} p_{ik} s_k$
13:         **end for**
14:         SET $\text{mi}_{mn} = \text{MI}\left(\left\{s_{i;m}^+\right\}_{i \neq m}, \{X_i\}_{i \neq m}\right)$
15:     **end for**
16:     SET $n^*(m) = \underset{n}{\operatorname{argmin}} \, \text{mi}_{mn}$
17:     SET $l_m = \lambda_{n^*(m)}$
18:     SET $s_i^+ = s_{i;i}^+$
19: **end procedure**
20: **procedure** INTERVAL PREDICTION
21:     SET $Q = \lceil(1-\alpha)(N+1)\rceil$.
22:     SET $p_{i(N+1)} = K_{i(N+1);i}/\sum_{k \neq i,N+1} K_{ik;i}$
23:     SET $\hat{\bar{s}}_{(N+1)i} = \sum_{k \neq i} p_{k(N+1)} s_k$
24:     SORT $S = [\hat{\bar{s}}_{(N+1)1} s_1^+, \ldots, \hat{\bar{s}}_{(N+1)N} s_N^+]$
25:     SET $\Delta y_{N+1} = S[Q]$
26: **end procedure**
**Ensure:** Test prediction interval $[m(X) - \Delta y_{N+1}, m(X) + \Delta y_{N+1}]$.

---

# B Dataset and Model Details

## B.1 One-Dimensional Regression

We formulate toy example of regression with label noise in one dimension as follows:

$$X \sim \mathrm{U}(0,1), \quad y = f_0 + X^2 \sin\left(k_f X + \phi_f\right) + \epsilon$$

$$\epsilon \sim \mathcal{N}(0, \nu(X)), \quad \nu(X) = \epsilon_0 + \left|\sin\left(k_\epsilon X + \phi_\epsilon\right)\right| \tag{8}$$

where

$$f_0 = 10^{-1}, \ k_f = 10, \ \phi_f = 1/2, \ \epsilon_0 = 10^{-2},$$

$$k_\epsilon = 2, \ \phi_\epsilon = 3 \times 10^{-1}, \ \lambda = 10^{-1}. \tag{9}$$

For each repetition of the experiment, we sample $i.i.d$ pairs of $(X, y)$ and split them into 1000 training points, 10000 test points and a variable number of calibration points. A random-forest regression model is trained, using the default parameters of SCIKIT-LEARN `v1.2.2` (Pedregosa et al., 2011) for our base model, or disabling bootstratpping to produce an overfitting model. For all methods, kernel scores are evaluated on the raw input values $X$

## B.2 Chemical Property Regression on TDC Datasets

We use two datasets from the Therapeutics Data Commons repository (Huang et al., 2021), each of which is treated independently. Both datasets contain samples consisting of pairs of SMILES (Weininger, 1988), descriptions of chemical structure as text sequences, and a target value for the property of interest. The datasets are already divided into training, validation and test samples, however, we merge the validation and test data and re-divide it randomly into calibration and test to allow for enough statistics. For each task, we fine tune a CHEMBERTA Chithrananda et al. (2020) language model pretrained on SMILES language modelling, provided by HUGGINGFACE.

The first task is to predict drug solubility on data originally produced by Sorkun et al. (2019) and contains 9982 samples, split into 6988 training points and 2994 test points.

The second task consists of estimating the drug microsome clearance (drug elimination rate by the liver) on 1102 samples from Hersey (2015), which are split into 772 training points and 330 test points.

Our fine-tuned CHEMBERTA models are defined HUGGINGFACE `AutoModelForSequenceClassification`[5] with `DeepChem/ChemBERTa-77M-MTR` weights and sequence data is processed with the adapted `AutoTokenizer`, with maximum sequence lengths set to the maximum sequence length in the training data. The training is performed with the mean-squared-error loss, using a batch size of 64 and a learning rate of $4.0 \times 10^{-5}$ over 100 epochs.

Embeddings for kernel scores are computed using the output of the `classifier.dense` layer of the model, which is the first linear layer of the classification head of the fine-tuned model.

## B.3 AlphaSeq Antibody Affinity Regression

We use the data from the AlphaSeq survey of 104,972 measurements of the binding affinity of antibody proteins to a SARS-CoV-2 target peptide, paired with amino-acid sequence information for the antibodies. We pre-process the label data by dropping missing measurements and measurements on reference epitopes (keeping only those labelled as `MIT_Target`), resulting in 69,297 valid sequence-affinity pairs. The data is divided into a training/validation/test+calibration split of sizes 39466/12517/17314. The test+calibration set is randomly subsampled into a test dataset of size 7314 and a calibration dataset of variable size.

---

[5]These models can also do regression, despite the name

We use the amino-acid sequences for both the heavy and the light chains of the antibody as inputs to pre-trained, frozen-weight AbLang Olsen et al. (2022) amino-acid language models adapted to each chain type. The embeddings thus produced are concatenated and mapped to numerical predictions with a two-hidden-layer neural network with layer widths $(128, 32)$ and `ReLU` activations. The final single-output layer does not have an activation function. This model is trained with the `Adam` optimizer using a learning rate of $1.0 \times 10^{-5}$, a batch size of 128 and is regularized with early stopping, monitoring the validation mean-squared error.

Embeddings for kernel scores are the concatenated outputs of the AbLang models.

### B.4  MNIST Regression

We use the classic MNIST dataset (LeCun & Cortes, 2005) repurposed as a regression task where each digit is labelled by its floating point value. We use data augmentation both at training time and at test time to increase the potential confusion between similar digits, which leads to a non-trivial error structure as we discuss in Appendix C.4.

We use the following randomized distortions, applied sequentially

- Gaussian blurring with kernel size $3 \times 3$, applied with probability 30%.

- Perspevtive transformation with scale 0.4, applied with probability 30% (`torchvision RandomPerspective`).

- Gaussian noise on each pixel with mean 0 and standard deviation $1/(6 + \nu)$ where $\nu \sim \mathrm{U}(0, 5)$.

The specific choice of transformation is quite arbitrary, but is meant to ensure qualitatively that numbers are nearly always recongnizable to human observers, while making significant distortions common. Transformations are resampled every time an image is used.

We train a convolutional neural network (CNN) on this regression task by optimizing the mean squared error loss with an ADAM optimizer with learning rate $5 \times 10^{-4}$. The CNN has the architecture described in Fig. 3.

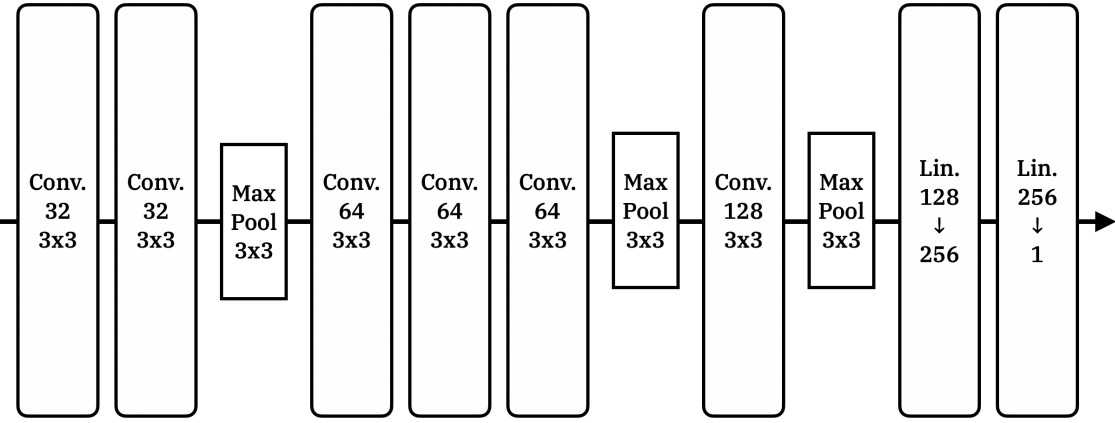

Figure 3: CNN used to perform the MNIST regression task. Each hidden layer is follwed by a batch normalisation layer and a rectified linear unit activation function.

Embeddings for kernel scores are the 256-dimensional outputs of the hidden linear layer.

### B.5  Hardware

All our experiments are run in a Linux HPC cluster. For model training and inference, we used 32-core, 64GB RAM environments with a single NVidia A100 GPU. Conformal predictions used the same CPU and RAM but no GPU. Experiments were run in Python 3.9 with pytorch v2.0.0+cu117.

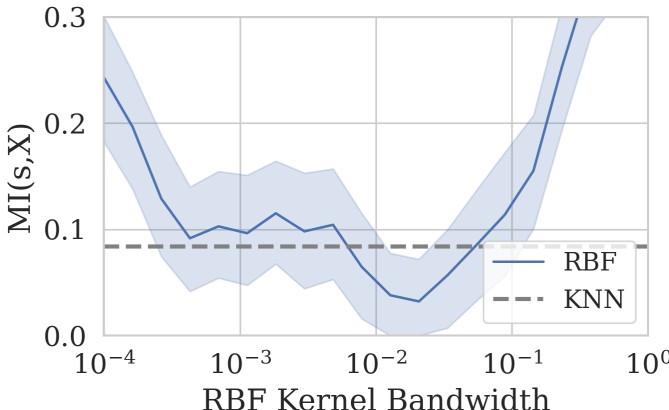

Figure 4: Dependence of the mutual information between $s^+(X, y)$ and $X$ as a function of the kernel length scale, reported as the average over each Jackknife+ split.

## C  Additional Experimental Results

### C.1  Original Kernel Tuning Strategies

We present in Table 3 an extended version of Table 1 including both our re-tuned kernels for the baselines, as described in Section 5.1 and the original tuning techniques proposed by each method. We report uncertainties as the $1 - \sigma$ percentile error, *i.e.* the $67^{\text{th}}$ percentile of absolute deviation from the mean, evaluated with bootstrapping. This is equivalent to reporting the standard deviation if the metrics are normally distributed, which is a weakly-motivated hypothesis given our limited statistics. As we describe in the body of the manuscript, the original tuning strategy proposed for LVD leads to many infinite intervals, sometimes a large majority. Given that it does reach reasonable performance in the 1D example, and that the worse case is ALPHASEQ, which uses very high-dimensional embeddings, we conjecture that this reflects the unsuitability of this tuning procedure in high dimensions.

In Table 4 we also present the same results for a set of datasets from the UCI repository that were examined in the paper (Lin et al., 2021).

### C.2  Jackknife+ Kernel Tuning

For for the sake of limiting computational costs and brevity, we demonstrated the performance of our method with a fixed 10-NN kernel. In this section, we compare this approach with a RBF approach, where the length scale is tuned using Algorithm 2, limiting ourselves to the one-dimensional example.

As we show in Table 5, a tuned RBF kernel yields comparable results to the KNN approach in the low statistics regime, but achieves better results when more data is available. As we use a Kozachenko-Leonenko-based mutual information estimator, the improved performance in the higher statistics regime is likely due its the asymptotically vanishing bias.

This potential for squeezing extra performance is confirmed by investigating the tuning procedure. In Fig. 4, we show that the is a finite range where $\text{MI}(s^+, X)$ is minimized, and that this minimizer does improve over the same measure evaluated on the 10-NN kernel.

### C.3  AlphaSeq Prediction Details

As we discuss in Section 5.3, the performance of the baselines is rather poor on the AlphaSeq dataset, which might be especially surprising for LVD due to the large calibration sets. The difficulties of MADSPLIT can

| Dataset | $N_{\text{cal}}$ | Method | Cov. $_{(\gtrsim 0.95)}$ | IS $^{\downarrow}$ | $R^2_{SQI}$ $^{\uparrow}$ | $\tau_{\text{SIQ}}$ $^{\uparrow}$ | $\tau_{\text{SI}}$ $^{\uparrow}$ |
|---|---|---|---|---|---|---|---|
| 1D | 500 | Ours | 0.96(1) | 0.18(1) | 0.87(3) | 0.91(4) | 0.35(1) |
| | | MADSPLIT (KNN) | 0.952(9) | 0.18(1) | 0.80(4) | 0.96(3) | 0.344(9) |
| | | MADSPLIT (Median) | 0.95(1) | 0.19(1) | $\ll 0$ | 0.1(1) | 0.04(1) |
| | | LVD (Tuned) | 0.982(3) | 0.21(1) | 0.61(9) | 0.86(4) | 0.33(1) |
| | | LVD (Base) | 0.985(2) | 0.22(1) | 0.5(1) | 0.79(9) | 0.31(2) |
| | 2000 | Ours | 0.952(2) | 0.171(5) | 0.90(2) | 0.96(4) | 0.36(1) |
| | | MADSPLIT | 0.951(3) | 0.184(2) | 0.81(2) | 0.93(6) | 0.340(7) |
| | | MADSPLIT (Median) | 0.95(1) | 0.19(1) | $\ll 0$ | 0.1(1) | 0.04(1) |
| | | LVD (Tuned) | 0.968(3) | 0.174(3) | 0.92(2) | 0.96(3) | 0.361(6) |
| | | LVD (Base) | 0.985(2) | 0.22(1) | 0.5(1) | 0.79(9) | 0.31(2) |
| TDC Sol. | 400 | Ours | 0.95(1) | 5.3(5) | 0.4(1) | 0.6(1) | 0.09(1) |
| | | MADSPLIT (KNN) | 0.95(1) | 5.0(3) | 0.53(5) | 0.71(6) | 0.182(3) |
| | | MADSPLIT (Median) | 0.95(1) | 4.6(3) | $\ll 0$ | 0.49(2) | 0.127(4) |
| | | LVD (Tuned) | 0.958(7) | 5.2(3) | 0.3(5) | 0.7(1) | 0.07(5) |
| | | LVD (Base) | 0.989(2) | 9(1) | $-0.8(2)$ | 0.6(1) | 0.06(1) |
| | 800 | Ours | 0.96(1) | 5.2(4) | 0.5(1) | 0.7(1) | 0.11(1) |
| | | MADSPLIT (KNN) | 0.951(6) | 5.1(2) | 0.58(7) | 0.71(5) | 0.179(6) |
| | | MADSPLIT (Median) | 0.95(1) | 4.7(2) | $\ll 0$ | 0.51(7) | 0.124(8) |
| | | LVD (Tuned) | 0.960(9) | 5.4(2) | 0.4(3) | 0.77(7) | 0.07(1) |
| | | LVD (Base) | 0.978(5) | 7.9(5) | $-0.1(1)$ | 0.8(1) | 0.058(7) |
| TDC Clear. | 60 | Ours | 0.95(2) | $1.9(3)\cdot10^2$ | 0.2(3) | 0.6(1) | 0.37(3) |
| | | MADSPLIT (KNN) | 0.95(2) | $1.9(5)\cdot10^2$ | 0.2(3) | 0.60(9) | 0.39(1) |
| | | MADSPLIT (Median) | 0.96(2) | $2.0(1)\cdot10^2$ | $\ll 0$ | 0.5(1) | 0.39(1) |
| | | LVD (Tuned) | 0.97(2) | $2.2(2)\cdot10^2$ | $\ll 0$ | $-0.3(6)$ | 0.1(2) |
| | | LVD (Base | 0.98(1) | $2.3(2)\cdot10^2$ | $\ll 0$ | $-0.0(3)$ | 0.00(5) |
| | 240 | Ours | 0.96(1) | $1.65(9)\cdot10^2$ | 0.3(2) | 0.5(1) | 0.39(7) |
| | | MADSPLIT (KNN) | 0.96(1) | $1.64(8)\cdot10^2$ | 0.41(5) | 0.6(1) | 0.41(7) |
| | | MADSPLIT (Median) | 0.96(1) | $1.94(2)\cdot10^2$ | $\ll 0$ | 0.50(7) | 0.41(5) |
| | | LVD (Tuned) | 0.97(1) | $2.02(5)\cdot10^2$ | $\ll 0$ | 0.1(2) | 0.1(1) |
| | | LVD (Base) | 0.995(6) | $2.57(3)\cdot10^2$ | $\ll 0$ | $-0.1(1)$ | $-0.03(4)$ |
| $\alpha$Seq CoVID | 2000 | Ours | 0.952(6) | 4.5(1) | 0.24(8) | 0.5(1) | 0.05(1) |
| | | MADSPLIT (KNN) | 0.952(5) | 4.6(1) | 0.1(2) | 0.2(2) | 0.01(2) |
| | | MADSPLIT (Median) | 0.950(5) | 4.06(7) | $\ll 0$ | 0.2(3) | 0.04(4) |
| | | LVD (Tuned) | 0.994(1) | 7.6(5) | $\ll 0$ | 0.1(1) | $-0.004(6)$ |
| | | LVD (Base) | 0.9997(3) | 8.6(8) | $\ll 0$ | 0.0(2) | $-0.003(6)$ |
| | 10000 | Ours | 0.953(4) | 4.54(6) | 0.28(6) | 0.6(1) | 0.052(8) |
| | | MADSPLIT (KNN) | 0.952(2) | 4.59(6) | 0.0(1) | 0.2(2) | 0.01(1) |
| | | MADSPLIT (Median) | 0.951(5) | 4.05(2) | $\ll 0$ | 0.1(4) | 0.01(7) |
| | | LVD (Tuned) | 0.973(4) | 5.49(4) | $-0.26(4)$ | 0.5(1) | 0.052(8) |
| | | LVD (Base) | 0.99993(6) | 9.54(3) | $\ll 0$ | 0.0(3) | $-0.001(9)$ |

Table 3: Conformal interval prediction performance metrics on N benchmark datasets. Numbers in parentheses are the uncertainty on the last significant digit evaluated over 10 random samples of test and calibration data.

| Dataset | $N_{\text{cal}}$ | Method | Cov. $_{(\gtrsim 0.95)}$ | IS $^\downarrow$ | $R^2_{SQI}$ $^\uparrow$ | $\tau_{\text{SIQ}}$ $^\uparrow$ | $\tau_{\text{SI}}$ $^\uparrow$ |
|---|---|---|---|---|---|---|---|
| UCI BostonHous. | 100 | Ours | 0.96(3) | $2.0(2)\cdot10^1$ | $-0.3(3)$ | 0.5(1) | 0.18(4) |
| | | MADSPLIT (KNN) | 0.94(2) | $1.8(1)\cdot10^1$ | $-0.4(7)$ | 0.5(1) | 0.20(4) |
| | | MADSPLIT (Median) | 0.95(3) | $1.9(2)\cdot10^1$ | $\ll 0$ | 0.0(1) | 0.05(4) |
| | | LVD (Tuned) | 0.97(2) | $2.1(3)\cdot10^1$ | $\ll 0$ | 0.2(4) | 0.17(3) |
| | | LVD (Base) | 0.990(9) | $3.7(6)\cdot10^1$ | $\ll 0$ | 0.1(1) | $-0.02(7)$ |
| | 150 | Ours | 0.95(2) | $2.0(1)\cdot10^1$ | $-1.0(9)$ | 0.4(1) | 0.19(7) |
| | | MADSPLIT | 0.94(3) | $1.8(1)\cdot10^1$ | $-1.3(12)$ | 0.4(1) | 0.20(7) |
| | | MADSPLIT (Median) | 0.95(2) | $1.9(1)\cdot10^1$ | $\ll 0$ | 0.0(1) | 0.06(8) |
| | | LVD (Tuned) | 0.96(3) | $2.0(1)\cdot10^1$ | $\ll 0$ | 0.3(2) | 0.19(6) |
| | | LVD (Base) | 0.99(1) | $3.9(6)\cdot10^1$ | $\ll 0$ | $-0.1(1)$ | $-0.04(8)$ |
| UCI Kin8nm | 600 | | 0.954(9) | 0.38(1) | 0.74(4) | 0.87(4) | 0.160(8) |
| | | MADSPLIT (KNN) | 0.949(5) | 0.364(9) | 0.81(1) | 0.88(5) | 0.205(3) |
| | | MADSPLIT (Median) | 0.949(7) | 0.37(1) | $\ll 0$ | 0.77(4) | 0.119(7) |
| | | LVD (Tuned) | 0.992(2) | 0.54(1) | $-0.7(3)$ | 0.63(7) | 0.098(7) |
| | | LVD (Base) | 0.997(1) | 0.65(3) | $\ll 0$ | 0.2(2) | 0.00(1) |
| | 2400 | Ours | 0.952(9) | 0.373(7) | 0.75(5) | 0.80(6) | 0.19(2) |
| | | MADSPLIT (KNN) | 0.948(8) | 0.364(4) | 0.76(4) | 0.78(4) | 0.21(1) |
| | | MADSPLIT (Median) | 0.950(6) | 0.370(3) | $\ll 0$ | 0.6(1) | 0.11(2) |
| | | LVD (Tuned) | 0.994(3) | 0.58(1) | $-0.8(2)$ | 0.5(1) | 0.08(1) |
| | | LVD (Base) | 0.9994(6) | 0.72(1) | $\ll 0$ | 0.1(1) | 0.00(2) |
| UCI BikeSharing | 200 | Ours | 0.95(1) | 0.58(5) | 0.79(6) | 0.8(1) | 0.19(1) |
| | | MADSPLIT (KNN) | 0.95(1) | 5.0(3) | 0.53(5) | 0.71(6) | 0.182(3) |
| | | MADSPLIT (Median) | 0.95(1) | 4.6(3) | $\ll 0$ | 0.49(2) | 0.127(4) |
| | | LVD (Tuned) | 0.96(1) | 0.59(5) | 0.6(2) | 0.93(6) | 0.202(4) |
| | | LVD (Base) | 0.97(1) | 0.73(9) | 0.5(2) | 0.87(7) | 0.18(1) |
| | 600 | Ours | 0.955(5) | 0.57(1) | 0.83(6) | 0.90(4) | 0.21(1) |
| | | MADSPLIT (KNN) | 0.952(8) | 0.51(2) | 0.69(6) | 0.90(3) | 0.286(2) |
| | | MADSPLIT (Median) | 0.951(9) | 0.59(3) | $\ll 0$ | 0.952(4) | 0.204(1) |
| | | LVD (Tuned) | 0.961(5) | 0.58(2) | 0.7(1) | 0.96(3) | 0.202(2) |
| | | LVD (Base) | 0.967(5) | 0.63(3) | 0.7(1) | 0.92(7) | 0.194(7) |
| UCI Concrete | 200 | Ours | 0.95(1) | $2.7(1)\cdot10^1$ | 0.2(2) | 0.4(1) | 0.15(3) |
| | | MADSPLIT (KNN) | 0.95(1) | 22.4(8) | 0.4(2) | 0.56(8) | 0.30(2) |
| | | MADSPLIT (Median) | 0.95(1) | 25.8(8) | $\ll 0$ | 0.46(9) | 0.15(3) |
| | | LVD (Tuned) | 0.979(7) | $3.2(3)\cdot10^1$ | $\ll 0$ | 0.41(8) | 0.08(4) |
| | | LVD (Base) | 0.994(6) | $4.7(10)\cdot10^1$ | $\ll 0$ | 0.1(1) | 0.00(6) |
| | 300 | Ours | 0.97(1) | $2.6(2)\cdot10^1$ | 0.2(2) | 0.6(1) | 0.20(4) |
| | | MADSPLIT (KNN) | 0.95(1) | $2.2(1)\cdot10^1$ | 0.3(2) | 0.5(1) | 0.30(5) |
| | | MADSPLIT (Median) | 0.96(1) | 26.2(4) | $\ll 0$ | 0.43(9) | 0.15(1) |
| | | LVD (Tuned) | 0.98(1) | $3.2(2)\cdot10^1$ | $-1.8(14)$ | 0.3(1) | 0.12(2) |
| | | LVD (Base) | 0.996(4) | $5.2(5)\cdot10^1$ | $\ll 0$ | $-0.0(3)$ | 0.00(3) |
| UCI Energy | 100 | Ours | 0.95(1) | $1.3(1)\cdot10^1$ | 0.4(2) | 0.6(1) | 0.33(3) |
| | | MADSPLIT (KNN) | 0.94(1) | 9.8(7) | 0.5(1) | 0.68(5) | 0.51(2) |
| | | MADSPLIT (Median) | 0.94(1) | $1.3(1)\cdot10^1$ | $\ll 0$ | 0.47(8) | 0.21(1) |
| | | LVD (Tuned) | 0.98(1) | $1.5(2)\cdot10^1$ | $\ll 0$ | 0.5(1) | 0.25(5) |
| | | LVD (Base) | 0.98(1) | $2.0(2)\cdot10^1$ | $\ll 0$ | $-0.1(1)$ | $-0.00(4)$ |
| | 200 | Ours | 0.94(4) | $1.1(1)\cdot10^1$ | 0.6(1) | 0.68(8) | 0.40(5) |
| | | MADSPLIT (KNN) | 0.96(2) | 10.2(8) | 0.7(1) | 0.7(1) | 0.54(6) |
| | | MADSPLIT (Median) | 0.94(3) | $1.3(1)\cdot10^1$ | $\ll 0$ | 0.48(9) | 0.23(5) |
| | | LVD (Tuned) | 0.96(2) | $1.3(1)\cdot10^1$ | $-0.6(5)$ | 0.4(1) | 0.27(8) |
| | | LVD (Base) | 0.99(1) | $2.0(1)\cdot10^1$ | $\ll 0$ | 0.1(1) | 0.05(5) |
| UCI Yacht | 50 | Ours | 0.97(2) | 10.0(2) | 0.8(1) | 0.91(4) | 0.64(2) |
| | | MADSPLIT (KNN) | 0.96(3) | 7.1(3) | 0.94(1) | 0.94(3) | 0.66(2) |
| | | MADSPLIT (Median) | 0.95(4) | $2.1(2)\cdot10^1$ | $\ll 0$ | 0.92(3) | 0.63(3) |
| | | LVD (Tuned) | 0.98(1) | $1.6(2)\cdot10^1$ | $\ll 0$ | 0.84(8) | 0.62(3) |
| | | LVD (Base) | 0.98(1) | $2.3(4)\cdot10^1$ | $\ll 0$ | 0.3(2) | 0.1(1) |
| | 90 | Ours | 0.94(2) | 7.9(3) | 0.7(1) | 0.82(8) | 0.6(1) |
| | | MADSPLIT (KNN) | 0.96(4) | 6.4(2) | 0.7(2) | 0.88(9) | 0.66(6) |
| | | MADSPLIT (Median) | 0.96(3) | 20.5(2) | $\ll 0$ | 0.82(9) | 0.6(1) |
| | | LVD (Tuned) | 0.98(1) | $1.2(1)\cdot10^1$ | $-0.2(2)$ | 0.8(1) | 0.6(1) |
| | | LVD (Base) | 0.98(1) | $1.5(1)\cdot10^1$ | $-0.3(3)$ | 0.5(1) | 0.5(1) |

Table 4: Same table as Table 3 for the UCI datasets examined in Lin et al. (2021).

| Dataset | $N_{\text{cal}}$ | Method | Cov. $_{(\gtrsim 0.95)}$ | IS $\downarrow$ | $R^2_{SQI}$ $^{\uparrow}$ | $\tau_{\text{SIQ}}$ $^{\uparrow}$ | $\tau_{\text{SI}}$ $^{\uparrow}$ |
|---|---|---|---|---|---|---|---|
| 1D | 500 | Ours (KNN) | 0.95(1) | 0.173(9) | 0.85(4) | 0.89(7) | 0.35(1) |
| | | Ours (tuned) | 0.95(1) | 0.17(1) | 0.84(8) | 0.85(6) | 0.34(2) |
| | 2000 | Ours (KNN) | 0.951(5) | 0.174(6) | 0.88(2) | 0.91(4) | 0.362(8) |
| | | Ours (tuned) | 0.949(3) | 0.157(4) | 0.93(1) | 0.92(4) | 0.37(1) |

Table 5: PI metrics evaluated on the 1-dimensional regression task described in Section 5.2, comparing our approach with a fixed 10-NN kernel and a RBF kernel tuned with Algorithm 2.

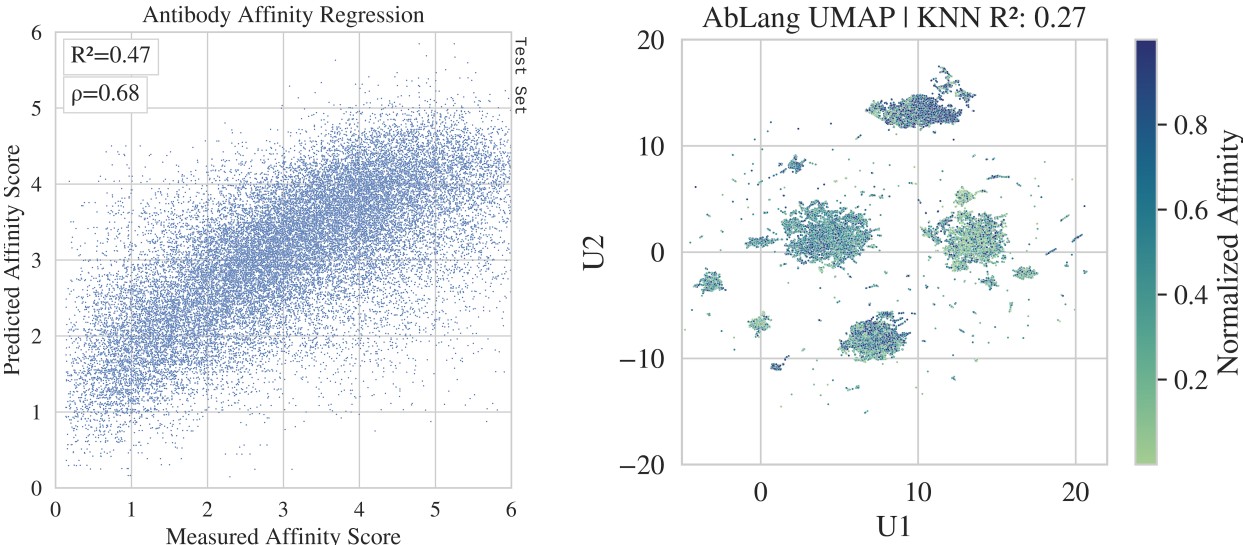

Figure 5: Visualization of our conjectured explanations for the difficulty of MADSPLIT and LVD on the ALPHASEQ dataset. (Left) Regression evaluation plot showing the systematic over and under prediction at the lower and higher edges of the label distribution. (Right) UMAP visualization of the ABLANG embeddings used for kernel similarity scores illustrating the scattered structure of the dataset.

be attributed to the clearly observable overfitting of our trained model, which shows in the form of an S-shape of the label-prediction plot of the test data, as seen in Fig. 5 (left). We attribute the degraded performance of LVD to the structure of data in the latent space used for similarity measurements: as we show in Fig. 5 (right), a UMAP of the test data highlights that despite much of the data being concentrated in large clusters, there is a significant number of isolated communities, which are particularly troublesome for LVD. Of course, UMAP representations can obfuscate many features of the data layout, and this explanation is only tentative.

## C.4   Analysis of the MNIST Regression Task Error Distribution

The MNIST Regression task stands out as the only one where no approach yields even a positive $R^2_{SQI}$, despite achieving rather high correlation scores (Table 1).

This observation can be explained by the reason that motivated our choice of this dataset, despite its seemingly contrived nature: the bulk and the tail of the per-label error distributions can be expected to be governed by two independent phenomena. On the one hand, most images are associated with a floating-point label that is distributed around the correct value[6], as can be seen in Fig. 6 (bottom), where the predictions for each digit have similar distributions close to the correct value. On the other hand, this same figure shows

---

[6]This spread is essentially dictated by the equilibrium reached between minimizing the MSE on clearly-identifiable examples, model capacity, and regularisation.

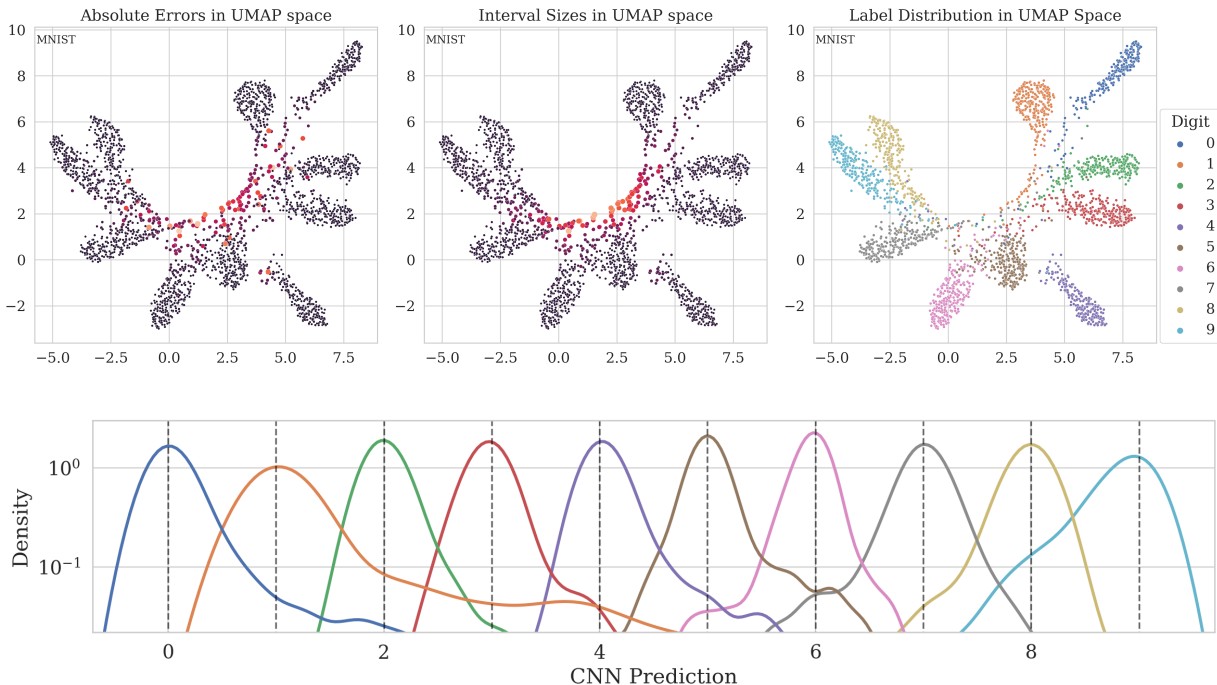

Figure 6: (Top) Latent-space structure for the MNIST Regression task interval sizes and labels visualized with UMAP. (Bottom) Prediction distribution per true label.

that the tail of the error distribution is very class-dependent. This tail structure is dictated by the confusion between digits induced by our test-time data augmentation, which can be observed in latent space, as shown in Fig. 6 (top): most of each class is well separated from the others, but each class has a tail overlapping with other classes, inducing high errors. Different digits have different fractions of confusing examples as well as different possible error classes, leading to the observed error dependence. As a result, our hypothesis that the conditional error quantiles is essentially some input-dependent rescaling of the conditional mean is badly broken. The error mean nevertheless captures some dependence of the error on the inputs, but the absolute interval size is thrown off, which is reflected in the high correlation, but low $R^2_{SQI}$.

## D    The Mean Interval Size is not a Sufficient Measure of Adaptivity

We argue that average interval sizes (IS) are a crude metric for measuring the adaptivity of a PI prediction method. Indeed, like any average-based metric, IS is sensitive to large outliers: a minority population with large errors would have little impact on the PI of a non adaptive method while a dynamic PI prediction would assign larger interval to this sub-population and lead to an increase in IS.

To illustrate this point Let us consider a regression dataset with $N$ elements whose regression errors follow a half normal distribution $|\mathcal{N}(0, \sigma)|$, where a fraction $\beta$ of points have scale factor $\sigma = \sigma_0$, while the other $(1 - \beta)$ have $\sigma = \lambda\sigma_0$. If $N$, $\lambda >> 1$ and $\beta > 0.95$, the standard conformal regression intervals based on absolute errors will be dictated by the normal errors and will therefore yield interval sizes close to twice the $95^{\text{th}}$ of the half-normal distribution:

$$\text{IS}_{\text{flat}} = 2 \times 1.96\sigma_0 \quad . \tag{10}$$

On the other hand, a perfectly adaptive PI predictor would yield this same $2 \times 1.96\sigma_0$ interval size for the $\sigma = \sigma_0$ population and $2 \times 1.96\lambda\sigma_0$ for the others. The average interval size of the "perfect" adaptive PI

would therefor be

$$\text{IS}_{\text{perfect}} = 2 \times 1.96\sigma_0 \left(\beta + (1-\beta)\lambda\right), \tag{11}$$

which will be larger than the flat average interval sizes for large enough $\lambda$.

We illustrate this point empirically in Fig. 7. As expected, the optimally adaptive PI increases when non-coverage risk $\alpha$ is comparable with the fraction of high error examples. This means that IS is inappropriate to use when the error distribution has long tail due to rare difficult examples, which is a common situation in image processing for example (Carlini et al., 2019; Baldock et al., 2021; Feldman, 2021), or due to skewed heteroscedatic errors.

(a) α=0.8                    (b) α=0.95

Figure 7: Average interval size ratio between a flat conformal and a perfectly adaptive PI when a errors are normally distributed, with $\sigma = \sigma_0$ for a fraction $\beta$, and $\sigma = \lambda\sigma_0$ for the remaining $1 - \beta$.

It is usually desirable to reduce IS at constant coverage, but this reduction is not necessarily a signal of improved adaptivity: the PI predictor might be systematically miscovering high-error minority population, favoring low-error minorities. This observation is what motivates our proposal for more correlation-oriented metrics in Section 2.2, which are sensitive to whether coverage distributes uniformly across high and low error populations.

## E  Formal Global Coverage Guarantees

Here we show how the conformal intervals defined in Eq. (7) provides the following theoretical guarantee:

$$P\left(Y_{N+1} \in C^{+\alpha}(X_{N+1})\right) \geq 1 - 2\alpha. \tag{12}$$

The proof is nearly identical to that provided in Section 6 of Barber et al. (2020b) and follows in four parts:

- We define a matrix of score competions between all the $\left\{s_{ij}^+\right\}_{i,j\in[1..N+1]}$, whose rows and columns are distributionally permutation-invariant.

- We use a theorem from Landau to show that there is an upper bound on the number of points that win atypically many competitions

- We use the distributional permutation invariance of the competition matrix to show that there is an upper bound on the probability that the test point $Z_{N+1}$ is such an "atypical winner".

- We show that $Y_{N+1} \notin C^{+\alpha}$ implies that $Z_{N+1}$ is an atypical winner, therefore obtaining an upper bound on the probability of this event by contraposition.

Let's get through each one

### E.1 Score and competition matrices

Let us define the following score matrix that has manifest distributional permutation invariance due to the exchangeability of $X_1, \ldots, X_{N+1}$:

$$R = (R_{ij}) = \begin{cases} s_{ij}^+ \text{ for } i \neq j \in [1..N+1] \\ R_{ii} = \infty \end{cases} \tag{13}$$

From this matrix, be further build a competition matrix

$$A_{ij} = \text{Indicator} (R_{ij} > R_{ji}). \tag{14}$$

Following the original proof, we define the set of "strange" points $S(A)$ as those that win abnormally many competitions:

$$S(A) = \left\{ i \in [1..N+1] \,\middle|\, \sum_j A_{ij} \geq (1-\alpha)(N+1) \right\}. \tag{15}$$

### E.2 Bounding strange points

There is a finite budget of victories in $A$ since $A_{ij} = 1 \Leftrightarrow A_{ji} = 0$, so there is an upper bound on the size of $S(A)$. This is formalized in a theorem from Landau (1953) that implies

$$|S(A)| \leq 2\alpha(N+1). \tag{16}$$

### E.3 From set sizes to probabilities

The matrix $R$, and therefore $A$ is distributionally permutation invariant, meaning that for any permutation matrix $\Pi$ and any possible matrix value $A_0$, $P(A = A_0) = P(A = \Pi A_0 \Pi^T)$. Permutations on the rows of $A$ correspond to permuting the $X_i$ so that

$$P(X_{N+1} \in S(A)) = P\left(X_j \in S\left(\Pi_{j,N+1} A \Pi_{j,N+1}^T\right)\right) = P(X_j \in S(A)), \tag{17}$$

where $\Pi_{j,N+1}$ is a permutation matrix exchanging rows $j$ and $N+1$. Therefore the probability that $N+1 \in S(A)$ is

$$\frac{\langle |S(A)| \rangle}{N+1} \leq 2\alpha, \tag{18}$$

where the inequality follows from the bound on $|S(A)|$.

#### E.3.1 Connecting strange points to coverage

Let us suppose that $X_{N+1}$ is not covered by its interval $C_\alpha(X_{N+1})$, which is defined as

$$C_\alpha = \{y \in \Omega_y \,|\, q_{1-\alpha}(\sigma_i(y)) \geq 1\}, \text{ where} \tag{19}$$

$$\sigma_i(y) = \frac{s_{i(N+1)}}{s(X_{N+1}, y, \theta_{i(N+1)})} \tag{20}$$

Non-coverage implies $q_{1-\alpha}(\sigma_i(y_{N+1})) < 1$ so that at least $\lceil (1-\alpha)(N+1) \rceil$ scores verify $\sigma_i(y_{N+1}) < 1$. This implies that that for at least $(1-\alpha)(N+1)$ indices $j \in [1..N]$, we have $s_{(N+1)j}^+ \geq s_{j(N+1)}^+$, thus making $X_{N+1}$ a strange point, which has probability bounded by $1 - 2\alpha$, therefore proving the coverage guarantee we announced in Proposition 3.

## F Method-Agnostic Results on Local Coverage

### F.1 Split-conformal conditional coverage guarantee

We argued in Section 2 that if inputs $X$ and scores $s(X, y)$ are independent random variables, then our weaker notion of conditional coverage guarantee, $A - \pi D$ coverage is realized. In practice however, the

independence of score and inputs will never be realized perfectly. We can nevertheless provide a bound on the local coverage based on their degree of independence, measured as a statistical distance between their joint and product distributions $p_{Xs}(X, s(X, y))$ and $p_X(X) \otimes p_s(s(X, y))$.

**Proposition 5** (Bound on conditional coverage)
*If the mutual information between $X$ and $s(X, y)$, $MI(X, s)$ is finite, for any $\omega_X \in \mathcal{F}_X$ such that $\mathbb{P}(X \in \omega_X) > \delta$*

$$\mathbb{P}(y \in S^\alpha(X) | X \in \omega_X) \geq (1 - \tilde{\alpha}), \qquad where \ \tilde{\alpha} = \alpha + \frac{\sqrt{1 - e^{-MI(X,s)}}}{\delta}.$$

This follows from the Bretagnolle-Huber theorem (see below) and the following lemma:

**Lemma 1** (main technical result)
*Let $p_{Xs}$ be the probability density of $(X, s(X, y))$ and $p_X$, $p_s$ the marginal densities of $X$ and $s(X, y)$. If $p_{Xs}(X, s)$ and $p_X(X) \otimes p_s(s)$ have finite total variation $\delta_{Xs}$, then on any $\omega_X \in \mathcal{F}_X$,*

$$\left| \mathbb{P}(y \in S^\alpha(X) | X \in \omega_X) - (1 - \tilde{\alpha}) \right| \times \mathbb{P}(X \in \omega_X) \leq \delta_{Xs}, \qquad where \ 1 - \tilde{\alpha} = \frac{\lceil (1 - \alpha)(n + 1) \rceil}{n + 1}. \tag{21}$$

**Bretagnolle-Huber Theorem**
*Given two proability distributions $P$ and $Q$ such that $P \ll Q$, then their total variation $\delta(P, Q)$ verifies*

$$\delta(P, Q) \leq \sqrt{1 - \exp(-D_{KL}(P||Q))}. \tag{22}$$

In particular, given two random variables $X, Y$, we have a bound on the total variation between their joint and product distributions expressed in terms of their mutual entropy:

$$\delta(p(X, Y), p(X)p(Y)) \leq \sqrt{1 - e^{-MI(X,Y)}}. \tag{23}$$

Let us now prove Lemma 1, which is the real meat of the result.

*Proof of Lemma 1.* Let $\omega_X \in \mathcal{F}_X$ such that $0 < |\omega_X| < \infty$. Let furthermore $q \in [0, 1]$ be the threshold defined from the conformal procedure with calibration set $\mathcal{C}_N$. We recall that marginally over $\mathcal{C}_N$, $\mathbb{P}(s(X, y) \leq q) \geq 1 - \alpha$.

Let us consider the following quantity, writing $s$ as shorthand for $s(X, y)$:

$$\Delta(\omega_X) = \left| \int_{\omega_X} dX \int_0^q ds \, p_{Xs}(X, s) - p_X(X)p_s(s) \right| \tag{24}$$

the integrand is the absolute difference of probabilities over the measurable set $\omega_X \times [0, q]$. By definition it is smaller that its supremum over all measurable sets; therefore

$$\Delta(\omega_X) \leq \delta_{Xs}. \tag{25}$$

Furthermore,

$$\Delta(\omega_X) = \left| \mathbb{P}(s \leq q, \, X \in \omega_X) - \mathbb{P}(s \leq q)\mathbb{P}(X \in \omega_X) \right|. \tag{26}$$

Therefore,

$$\mathbb{P}(s \leq q | X \in \omega_X) \mathbb{P}(X \in \omega_X) \geq (1 - \alpha)\mathbb{P}(X \in \omega_X) - \delta_{Xs}. \tag{27}$$

$\square$

The bound becomes vacuous on very unlikely sets, which seems unavoidable. This limitation is related to the fact that $\delta$ or MI are global measures and that the contribution from any small set is small. Therefore, a small set can deviate from the mean by a factor that is inversely proportional to its probability. Nevertheless, decreasing the mutual information uniformly decreases the penalty in the local coverage. We have furthermore observed empirically that minimizing the score-input mutual information for our method improves the PI evaluation metrics, even when the bound derived from the minimum is very weak.

### F.2   Jackknife+ conditional coverage guarantee

The proof for Proposition 4 is nearly identical to that of Proposition 2, as the very similar formulation suggests. Let us start by repeating the result.

**Proposition 6**
*Consider the data in Section 1.1, the score matrix $s_{ij}$ defined above and define $k_\alpha = \lceil (1-\alpha)(N+1) \rceil$. We define the score ratio $\sigma$ as the $k_\alpha$-th among the sorted score ratios $s_{(N+1)i}/s_{i(N+1)}$. Then for any $\omega_X \in \mathcal{F}_X$ such that $\mathbb{P}(X \in \omega_X) \geq \delta$*

$$\mathbb{P}\left(y_{N+1} \in C_\alpha(X_{N+1})|X \in \omega_X\right) \geq 1 - \alpha", \quad \text{where } \alpha" = 2\alpha + \frac{\sqrt{1 - e^{-MI(X,\sigma)}}}{\delta}.$$

Indeed, the definition of $C_\alpha$ in Eq. (3) is based on the value of an empirical quantile of the score ratios $s_{(N+1)i}/s_{i(N+1)}$, which is exactly the $k_\alpha$-th order statistic in this set of $N$ ratios. The global coverage guarantee ensures therefore that

$$\mathbb{P}(\sigma \leq 1) \geq 1 - 2\alpha. \tag{28}$$

Defining the $\Delta(\omega_X)$ as in Eq. (24) but replacing $s$ with $\sigma$ and $q$ by 1, we find the analog of Eq. (26):

$$\Delta(\omega_X) \leq \delta_{X\sigma}, \quad \text{and} \quad \Delta(\omega_X) = \left| \mathbb{P}\left(\sigma \leq 1 | X \in \omega_X\right) - \mathbb{P}(\sigma \leq 1) \right| \mathbb{P}(X \in \omega_X), \tag{29}$$

which establishes

$$\mathbb{P}\left(\sigma \leq 1 | X \in \omega_X\right) \mathbb{P}(X \in \omega_X) \geq (1 - 2\alpha)\mathbb{P}(X \in \omega_X) - \delta_{X\sigma}. \tag{30}$$

# G    Reproducibility Checklist

In this section, we provide a reproducibility checklist.

### G.1    General

This paper:

**Includes a conceptual outline and/or pseudocode description of AI methods introduced**    Yes. We describe the methods in plain text in Sections 3 and 4 and provide algorithms in the supplementary materials.

**Clearly delineates statements that are opinions, hypothesis, and speculation from objective facts and results**    Yes

**Provides well marked pedagogical references for less-familiar readers to gain background necessary to replicate the paper**    Yes. We include a "related work" section that refer to foundational references.

### G.2    Theory

**Does this paper make theoretical contributions?**    Yes

**All assumptions and restrictions are stated clearly and formally.**    Yes.
The general setup is delineated in the Definitions section and theoretical results start with further assumptions.

**All novel claims are stated formally (e.g., in theorem statements)**    Yes.

**Proofs of all novel claims are included.**    Yes, in the supplementary material.

**Proof sketches or intuitions are given for complex and/or novel results.**    Yes.
The proofs are structured into steps that make it easy to follow the flow without the details.

**Appropriate citations to theoretical tools used are given.**    Yes.

**All theoretical claims are demonstrated empirically to hold.**    Yes.

**All experimental code used to eliminate or disprove claims is included.**    Yes

### G.3    Datasets

**Does this paper rely on one or more datasets?**    Yes.

**A motivation is given for why the experiments are conducted on the selected datasets**    Yes.

**All novel datasets introduced in this paper are included in a data appendix.**    N/A.

**All datasets drawn from the existing literature (potentially including authors' own previously published work) are accompanied by appropriate citations.**    Yes.

**All datasets that are not publicly available are described in detail, with explanation why publicly available alternatives are not scientifically satisficing.**    N/A.

### G.4 Experiments

**Does this paper include computational experiments?**   Yes.

**Any code required for pre-processing data is included in the appendix.**   No.
Code will be released with a permissive open-source licence upon publication.

**All source code required for conducting and analyzing the experiments is included in a code appendix.**   No.
Code will be released with a permissive open-source licence upon publication.

**All source code required for conducting and analyzing the experiments will be made publicly available upon publication of the paper with a license that allows free usage for research purposes.**   Yes.

**All source code implementing new methods have comments detailing the implementation, with references to the paper where each step comes from.**   Yes.
Our code separates cleanly the method from experiments to allow reuse of the method and an effort was given to organization and clarity.

**If an algorithm depends on randomness, then the method used for setting seeds is described in a way sufficient to allow replication of results.**   Yes.
All experiments on our and benchmark methods use explicit seeds for repetitions, which are listed explicitly in the experimental code. Models were trained once since their performance was not our focus and weights will be released with the rest of the code.

**This paper specifies the computing infrastructure used for running experiments (hardware and software), including GPU/CPU models; amount of memory; operating system; names and versions of relevant software libraries and frameworks.**   Yes

**This paper formally describes evaluation metrics used and explains the motivation for choosing these metrics.**   Yes

**Analysis of experiments goes beyond single-dimensional summaries of performance (e.g., average; median) to include measures of variation, confidence, or other distributional information.**   Yes

**The significance of any improvement or decrease in performance is judged using appropriate statistical tests.**   Yes

**This paper lists all final (hyper-)parameters used for each model/algorithm in the paper's experiments.**   Yes.
Again we specify which (few) hyperparameters were used for the CP methods used. Details of the models are not relevant to our results but their weights will be released.

**This paper states the number and range of values tried per (hyper-) parameter during development of the paper, along with the criterion used for selecting the final parameter setting.**   N/A.

