# OpenReview forum: "Adaptive Conformal Regression with Split-Jackknife+ Scores"
_TMLR — Accepted by TMLR_

### Review · Reviewer_TpGN · 2024-01-24

**Summary Of Contributions:**

The paper proposes a new method for conformal prediction in regression to produce dynamically sized prediction intervals. The proposed method combines ideas from the split-CP and Jackknife+ methods. Empirical validation shows improved performance when the predictive model has a lot of overfitting.

**Audience:**

Yes

**Broader Impact Concerns:**

I have no ethical concerns regarding this paper.

**Claims And Evidence:**

Yes

**Requested Changes:**

Please fix the problems highlighted above as weaknesses.

**Strengths And Weaknesses:**

Strengths:

The paper proposes a new method and provides both theoretical and empirical results.

The empirical results show improved performance of the proposed method when the predictive model has a lot of overfitting.

The paper is mostly well-written, except for the below listed weaknesses.

Weaknesses:

The paper does not explain the existing literature to a sufficient extent. In particular, it lacks references to papers on the calibration of probabilistic regression methods, such as by [Kuleshov et al 2018] and several newer works published after that. While those methods do not provide the same kind of guarantees as conformal prediction, they are still very important in solving the same task in practice. Thus, it would be important to explain to the reader what the similarities and differences are.

In Section 5.3, just before Section 5.3.1 starts, it has not been explained what is meant by the "ideal absolute interval size". In Fig. 2, the evaluation measure used for the y-axis has not been defined and its choice has not been justified.

Evaluation measures used in Table 1 have not been appropriately defined. Also, it has not been explained why these particular measures are the best ones to use in the context of this paper.

Some more notation and terminology have not been appropriately defined:
page 2: s_i
page 7, Section 5.1, first line - 'localizer'

Page 5, last paragraph: Notation is confusing when defining $T_{ij}$ as $i$ is used in two different meanings: $i\in J^+ \setminus\{i,j\}$. Please use another name for the bound variable.

In Section 5.2, is the synthetic 1D regression problem an original one or found from the literature, possibly with minor modifications? If it is from the literature, then please return to the original. If it is original, then please briefly justify what the desiderata are behind the choice of this particular function (can be a very brief justification).

On page 9, there is insufficient explanation about what the 'floating-point prediction of labels of MNIST images' means. Although there is a reference to the original paper, it would be good to explain it at least in 1-2 sentences.

On page 9, there are some typos and errors:
amouts -> amounts
'statistics highlight confirm' - remove either 'highlight' or 'confirm'.
'both method' -> 'both methods'

In the appendices, Algorithms 1 and 2 have some undefined variables.
In Algorithm 1, s_k is undefined and the definition of $K_{ij}$ is a bit strangely pushed to the right margin.
Analogical problems occur in Algorithm 2.

[Kuleshov et al 2018] Kuleshov, V., Fenner, N. and Ermon, S., 2018, July. Accurate uncertainties for deep learning using calibrated regression. In International conference on machine learning (pp. 2796-2804). PMLR.
https://proceedings.mlr.press/v80/kuleshov18a/kuleshov18a.pdf

---

> ### Author Response · Authors · 2024-02-25
> **Response to reviewer TpGN**
>
> Many thanks to reviewer TpGN for their evaluation of our manuscript and the points they raised as requiring improvement. We have tried to address the highlighted weaknesses in the updated version of our text, with changes highlighted in blue and provide a summary below
> - We appreciate your request for placing our work in a broader context of uncertainty quantification in ML, we have expanded the introduction to this effect.
> - We have replaced 'ideal interval sizes' with 'adaptive and tight intervals' and expressed why this is measured by $R^2_\text{SQI}$ in section 2.2.3
> - Section 2.2 introduces the metrics we use in table 1. and explains why they are adapted. There was a typo in the name of $R^2_\text{SQI}$, which we corrected.
> - The term localizer was used synonymously with kernels, we removed its uses.
> - We corrected the issue with the definition of $T_{ij}$
> - The relationship used for the 1D example is original and was chosen somewhat arbitrarily with the requirement that it would have a non-constant rate of change and heteroscedatic noise out of phase with the function. We made this explicit below the definition. The oscillatory behavior of the function was chosen for aesthetic considerations
> - We have rephrased the definition of the MNIST task (where we perform regression on the numeric label of MNIST images, i.e. predict the number "5.0" if the image is a 5). This is a contrived example, but well-suited to highlighting a failure mode common to all methods as we discuss in the appendix.
> - We have addressed the issue with symbol definitions in the algorithm

---

### Review · Reviewer_XjJj · 2024-01-26

**Summary Of Contributions:**

The authors propose a conformal regression method designed to generate dynamically-sized prediction intervals. The proposed approach combines the split CP and Jackknife+ procedures to fine-tune score functions using calibration data. The motivation for this method is grounded in distribution-dependent conditional coverage guarantees that consider the statistical dependence between the input variable and the scores. Specifically, the goal is to improve the conditional validity of conformal prediction sets by reducing this dependence through adapting the score function to the data distribution. The proposed method demonstrates greater robustness to overfitting than the MADSplit method and is more sample-efficient than recent ECDF-based methods.

**Audience:**

Yes

**Broader Impact Concerns:**

NA.

**Claims And Evidence:**

No

**Requested Changes:**

- **Clarification Needed:**
	- The statement "multiple notions of conditional coverage guarantees are impossible," is unclear. It should be noted that while conditional coverage is impossible to achieve in a finite-sample context, various other notions of approximate conditional coverage have been proposed.
	- Definition 3: The expression "if there is" needs clarification. Why is mere existence sufficient?
	- Proposition 2: Are there examples of scores that exhibit the described property? The authors should discuss this in more detail, perhaps with examples like the HPD-split method (see Section 3.2 in CD-split and HPD-split: Efficient Conformal Regions in High Dimensions). This paper should also be discussed in the related work.
	- Page 5: The definition of T_ij is confusing and needs clarification.
	- The term "a sufficiently small penalty coefficient" is vague. What does "penalty coefficient" refer to?
	- Clarify what is meant by "errors" in the context of "A natural metric to measure whether errors and interval sizes."
	- The term "quantile aggregation" could be interpreted differently depending on the context. More specificity is required.
	- The sentence regarding a problem-dependent guarantee similar to Theorem 3 is unclear. More details should be provided or the sentence should be removed.
	- Proposition 5: Clarify whether "Consider the data" refers to specific notations.
	- Clarify why "The use of Theorem 3 is a posteriori."


- **Improvement Needed:**
	- Sections 2.2.2 and 2.2.3 are quite confusing. A more formalized approach and detailed explanation of the computed quantities are needed.
	- In Section 2.1, when discussing "abandoning distribution-independence," the implications of this assumption should be explored. Are the methods still within the conformal framework? The authors should elaborate on this point or provide examples of other conformal methods that incorporate distributional assumptions.
	- The transition from Section 3.1 to 3.2 is abrupt and needs smoothing. Further, the explanations regarding sigma and the embedding in (2) lack detail and appear arbitrary. This being the objective function, clarification is essential.
	- Discuss the fairness of "showing that rescaled scores yield a significant reduction in the score-input mutual information compared to the raw absolute error score," especially since mutual information is included in the objective function.


- **Experiments:**
	- The authors are advised to include datasets from MADSplit (Lei et al., 2017) and LVD (Lin et al., 2021) in their experiments, as these are the methods their own method is being compared to.
	- It is recommended not to exclude the SLCP baseline from the study, even if it does not achieve comparable accuracy.
	- Section 2.2: Other works have observed similar phenomena. Consider discussing the use of geometric means, i.e., averaging log values.
	- While the authors highlight the difficulty of parameter tuning in other conformal methods, they should address the parameter tuning (using K=10) in their own method.
	- Explain why "a large fraction of infinite intervals" occurs, possibly adding more details in the appendix.
	- Table 2: Discuss the sensitivity of the procedure to the chosen delta value (0.3).


- **Inconsistent Terminology:**
	- "Sets of prediction" might be more accurately termed "prediction sets."
	- The term "any desired expected coverage" could be clarified. In Section 1.1, the term "marginal risk" is used, whereas "a global coverage guarantee" is mentioned in Section 1.2.1.
	- "The right answer" could be more precisely referred to as "the realization."
	- ...


- **Errors:**
	- Theorem 7 should be corrected to Lemma 7.
	- Theorem 3 should be corrected to Proposition 3.
	- MADSplit is mentioned in the abstract and introduction without adequate context or introduction.
	- "We confirm this intuition empirically in REF" requires a specific reference.
	- In Section 1.2.1, ensure consistency with C_N as defined in Section 1.1.
	- Section 1.2.3: Consider mentioning overfitting in the context of "poor performance when errors differ between training and test sets."

**Strengths And Weaknesses:**

- **Strengths:**
    - The paper addresses the significant issue of enhancing conditional coverage in conformal prediction.
    - It innovatively combines two well-established conformal methods, split conformal and jackknife+, to improve conditional coverage in regression problems.

- **Weaknesses:**
    - The paper's major flaw is its substandard writing quality and inconsistent use of notation and terminology, necessitating substantial revisions for clarity.
    - The authors modify the distribution-free assumption fundamental to conformal prediction. The paper does not adequately discuss the consequences of this change, nor does it transparently compare its fairness with other methods.
    - While numerous conformal regression methods exist, the paper limits its comparison to only two other methods.
    - The authors overlook the datasets used in MADSplit (Lei et al., 2017) and LVD (Lin et al., 2021), their two baseline methods.

---

> ### Author Response · Authors · 2024-02-25
> **Response to reviewer XjJj**
>
> We thank reviewer XjJj for their careful review of our manuscript and for their recommendations for improvements. We have uploaded a revised manuscript with our changes highlighted in blue.
>
> Clarifications/Improvements:
> - We have clarified our statement that there are several impossibility theorems for finite-sample conditional coverage.
> - Definition 3 is weak but is formulated as intended and prepares the ground for proposition 3: we consider prediction sets with a marginal coverage guarantee of $1-\alpha$ and obtain a weak conditional guarantee of $1-\alpha'$ when conditioning on sets with probability at least $\delta$
> - We don't believe there is a score function that verifies Proposition 2 in general. Proposition 2 serves as a pedagogical tool to understand proposition 3 in the limiting case $\text{MI}(X,s)=0$.
> - Thank you for pointing out the inconsistent notation, the definition of $T_{ij}$ is now fixed.
> - We have clarified how the guarantee of Proposition 3 is the global guarantee $1-\alpha$ degraded with a penalty term: $-\sqrt{1-e^{\text{MI(X,s)}}}/\delta$, which should help the discussion of when the bound becomes vacuous.
> - We have hopefully significantly improved the introduction of section 2.2 and the definition of the metrics.
> - There was an issue in the numbering of Theorems/Propositions/Lemmas which should now be fix and clear some inconsistencies.
> - The phrase "The use of theorem 3 is a posteriori" has been removed and replaced with a discussion of the distribution- and score-dependence of the modified risk $\tilde \alpha$. We further added the following comments which we think important to respond to the question of whether we are still working within the framework of conformal predictions:
> "While the conditional risk $\tilde \alpha$ is problem-dependent, Proposition 3 is still a result about conformal predictions in that it characterizes their conditional properties. Proposition 3 applies generally to split-conformal predictions, for final samples and without making hypotheses about classes of distributions."
> - We have improved the transition between sections 3.1 and 3.2
>
>
> Experiments
> - We have added Table 4 which reproduces the tabular datasets used in the MADSplit and LVD papers.
> - We added a footnote explaining the appearance of infinite intervals in LVD. This is a feature of ECDF-reweighting methods, which is arguably reasonable given adapted kernels (test samples very different from the calibration data yield very uncertain predictions). However our findings indicate that the tuning method proposed for LVD doesn't.
> - We simply did not tune our procedure: we chose K=10 as a number small enough for KNN calculations to be reasonably fast in our simulations and kept it constant.

---

> > ### Comment · Reviewer_XjJj · 2024-03-08
> >
> > Thank you for addressing my comments. The paper still needs improvement in terms of clarity and writing. Here are some additional minor comments:
> >
> > - Please clarify this sentence: "- is slow-varying and wide enough that predicting conditional quantiles is a valid learning objective"
> > - Page 4, What is ApiC? Do you mean ApiDC?
> > - The authors write "coverage is equivalent to AE <= |I|/2". How does that relate to expression (1)?
> > - "which is obtained if "x I_d"". What is "x I_d"?
> > - Typos
> >   -  We introduce a further weakened notion conditional coverage"
> >   - Page 27: "four our method"
> >    - Page 5:  "final samples"

---

> > > ### Author Response · Authors · 2024-03-11
> > >
> > > Many thanks for finding these imprecisions and typos, which we have tried to address as follows:
> > > - We have removed the imprecise statement about slow-varying p(y|X). We were trying to express the intuition that estimating quantiles of p(y|X) for quantile regression often requires more data than estimating expectation values such as MADSplit. However, we did not find suitable references to support that statement.
> > > - We updated the manuscript to detail a connection between coverage and equation (1), which we hope will be satisfactory.
> > > - Indeed ApiC was a typo for ApiDC
> > > - the "x" symbol in "x Id" was also a typo
> > > - We have fixed the three further typos you noticed, thank you for pointing them out.

---

### Review · Reviewer_cwzg · 2024-02-03

**Summary Of Contributions:**

Conformal prediction is a framework to obtain post-hoc uncertainty quantification for probabilistic machine learning methods based on a hold-out calibration datasets.
The paper introduces a new method that extends the popular Jackknife+ procedure for producing dynamically-sized prediction intervals in regression problems.
The proposed method performs empirically on-par or better than baseline methods, for example, when the size of the calibration dataset is small.

**Audience:**

Yes

**Claims And Evidence:**

Yes

**Requested Changes:**

The introduction could provide a better motivation of the proposed approach. If possible, a figure that explains the overall approach would help.

### Minor fixes:

- Figure 1: Missing labels that indicates the different lines in the plot.
- Figure 2: What do you actually show on the y-axis?  R_{SQI} is not defined?
- What do bold entries mean in Table 1

**Strengths And Weaknesses:**

## Strengths

- The proposed method seems reasonable and extends the current literature

- Overall, I found the paper to be well written.


## Weaknesses

- The paper could benefit from a more gentle introduction into conformal prediction and a better motivation of the proposed approach in the introduction session.  This would make paper easier to understand for readers (like me) that are not deeply familiar with the topic

- The paper includes a lot of jargon, which would be probably fine for a conference paper, but for a journal article it might help to better introduce these terms such that the paper becomes more self-contained.

- In intro, the paper says that the proposed method performs better than MADSplit in case of overfitting. But I am not sure how this is reflected in the empirical evaluation of the method. For example, Figure 2 only shows the size of the calibration set and the performance of MADSplit seems to be constant here.

---

> ### Author Response · Authors · 2024-02-25
> **Response to reviewer cwzg**
>
> We thank reviewer cwzg for their evaluation of our manuscript and their requests for improvements. We have uploaded a revised manuscript with our changes highlighted in blue.
>
> - In the new version of the manuscript, we propose an expanded introduction that better motivates our work.
> - We might indeed have skimmed too fast over background in conformal predictions as well as jargon. We would be happy to provide an explanation of the jargon but would appreciate it if you could please point us to which terms need to be translated. On the other hand, we have introduced ourselves to the general domain of CP with the excellent resource: https://arxiv.org/abs/2107.07511. We would likely wind up rephrasing much of that source. Would it be suitable to cite this resource prominently in the introduction or background section?
> - With regard to overfitting with MADSplit, we do indeed not illustrate how overfitting affects the performance directly, as changing the size of the calibration set does not change the generalization of the model. Figure 2 illustrates, however, that there is an irreducible performance ceiling for MADSplit, whose R2 does not increase with more calibration data, while both our method and the other method do. As the difference between our method and MADSplit is the data on which the local error mean estimator is trained (training vs calibration data), we attribute the difference in performance to the different error distributions between training and calibration data.
>
> Minor fixes:
> - There was a typo in our metric definitions; $R^2_\text{SQI}$ is now defined.
> - We specify the meanings of the different lines in the caption of figure 1
> - We added an explanation of the bold entries in the caption of table 1

---

### Decision · Action_Editor_SxWC · 2024-03-11

**Recommendation:** Accept as is

**Comment:**

The paper proposes a new method for conformal prediction that extends the popular Jackknife+ procedure for producing dynamically-sized prediction intervals in regression problems. Improvements over existing baselines are shown in small-data regimes and shows better robustness to overfitting.

**Audience:**

The submission addresses the important problem of improving the conditional validity of conformal prediction sets, and would be of interest to TMLR's audience.

**Claims And Evidence:**

After revisions as requested by the reviewers, the submission is now ready to be published.